# Learning About Progress From Experts

**Jake Bruce**
DeepMind

**Ankit Anand**
DeepMind

**Bogdan Mazoure**[*]
McGill University

**Rob Fergus**
DeepMind

## Abstract

Many important tasks involve some notion of long-term progress in multiple phases: e.g. to clean a shelf it must be cleared of items, cleaning products applied, and then the items placed back on the shelf. In this work, we explore the use of expert demonstrations in long-horizon tasks to learn a monotonically increasing function that summarizes progress. This function can then be used to aid agent exploration in environments with sparse rewards. As a case study we consider the NetHack environment, which requires long-term progress at a variety of scales and is far from being solved by existing approaches. In this environment, we demonstrate that by learning a model of long-term progress from expert data containing only observations, we can achieve efficient exploration in challenging sparse tasks, well beyond what is possible with current state-of-the-art approaches. We have made the curated gameplay dataset used in this work available at `https://github.com/deepmind/nao_top10`.

## 1 Introduction

Complex real-world tasks often involve long time dependencies, and require decision making across multiple phases of progress. This class of problems can be challenging to solve due to the fact that the effects of multiple decisions are intertwined together across timesteps. Moreover, the sparsity of the learning signal in many tasks can result in a challenging exploration problem, which motivates the use of intrinsic sources of feedback, e.g. curiosity (Pathak et al., 2017; Raileanu & Rocktäschel, 2020), information-gathering (Kim et al., 2018; Ecoffet et al., 2019; Guo et al., 2022), diversity-seeking (Hong et al., 2018; Seo et al., 2021; Yarats et al., 2021) and many others.

The internal structure of some environments implicitly advantages certain types of intrinsic motivation. For example, Go-Explore (Ecoffet et al., 2019) excels on Montezuma's Revenge by enforcing spatio-temporally consistent exploration, while ProtoRL (Yarats et al., 2021) achieves high returns in continuous control domains. Nevertheless, some challenging tasks that do not have this structure remain unsolved due to complex dynamics and large action spaces. For instance, in this work we study the game of NetHack (Küttler et al., 2020), which manifests a mixture of long task horizon, sparse or uninformative learning signal and complex dynamics, making it an ideal testbed for building exploration-oriented agents. The complexity of NetHack prevents agents from efficiently exploring the action space, and even computing meaningful curiosity objectives can be challenging.

Instead of training agents on NetHack *tabula rasa*, a challenging prospect for efficient exploration, we take inspiration from recent advances in another hard exploration benchmark, Minecraft (Baker et al., 2022; Fan et al., 2022), and leverage plentiful offline human demonstration data available in the wild. Equipping learning agents with human priors has successfully been done in the context of deep reinforcement learning (Silver et al., 2016; Cruz Jr et al., 2017; Abramson et al., 2021; Shah et al., 2021; Baker et al., 2022; Fan et al., 2022) and robotic manipulation (Mandlekar et al., 2021).

One of the salient features of domains such as NetHack (Küttler et al., 2020) is the lack of an explicit signal for monotonic progress. The objective of the game is to descend through the dungeon obtaining equipment, finding food & gold, battling monsters and collecting several critical items along the way, before returning to the beginning of the game with one particular item. A full game can take even experienced players upward of 24 hours of real time and more than 30k actions, and the game can be punishingly difficult for all but the best players. In addition, the in-game score is not a meaningful measure of progress, and maximizing it does not lead agents toward completing the game (Küttler et al., 2020). Correspondingly, agents trained via reinforcement learning methods fail to make significant long-term progress (Hambro et al., 2022a).

---

[*]This work was done while the author was doing internship at DeepMind. Author is now at Apple.

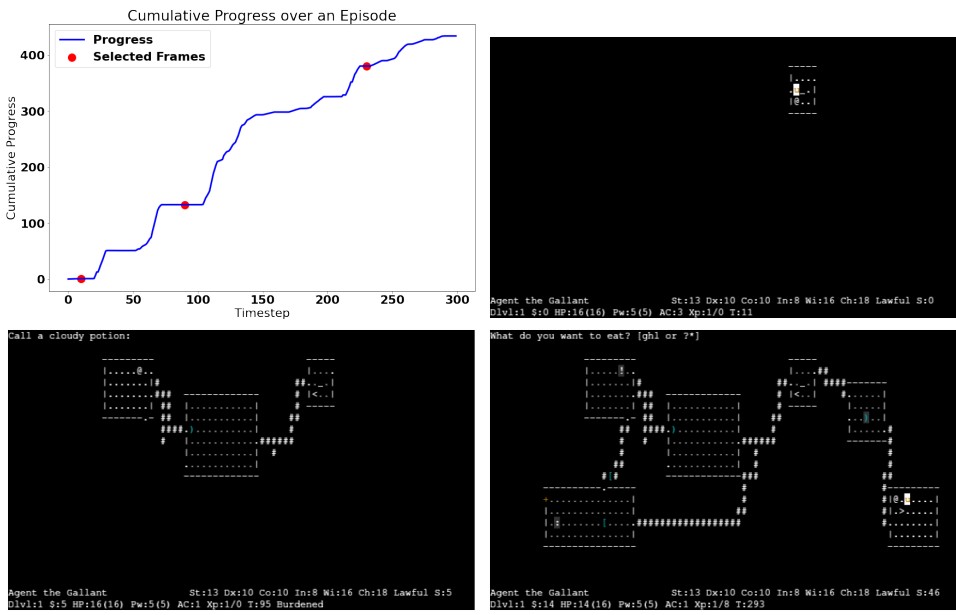

Figure 1: In this work, we learn a model of task progress from expert demonstrations and use it as an exploration reward. This figure shows an example cumulative progress curve over the first 300 steps of a representative episode, alongside three frames taken over the course of the sequence. In the course of a typical episode, the player @ will explore a procedurally-generated dungeon, interact with objects ) [ , open locked and unlocked doors + search for the stairs to the next level > , and more.

Many tasks of interest combine long horizons, complex temporal dependencies and sparse feedback, but demonstrations with actions are often difficult to obtain: we focus on these settings where abundant unlabeled data is available in the wild. To address these problems we propose Explore Like Experts (ELE), a simple way to use progress estimates from expert demonstrations as an exploration reward. We estimate progress by training a model to predict the temporal distance between any two observations; the model is pre-trained on offline human demonstrations and does not require actions. Maximizing an exploration objective based on this progress model leads to new state-of-the-art performance on complex NetHack tasks known for their difficulty, including four tasks on which other competing methods achieve no reward at all. While we focus on NetHack in this work, the method is not specific to that domain, and can in principle be applied to any challenging exploration task with demonstration data that incorporates a notion of long-term progress.

## 2 RELATED WORKS

**Intrinsic motivation:** In complex domains with sparse extrinsic feedback, agents may resort to intrinsic motivation. Existing intrinsic motivation objectives can be informally categorized into curiosity-driven (Pathak et al., 2017; Zhang et al., 2020; Raileanu & Rocktäschel, 2020), information-gathering (Kim et al., 2018; Ecoffet et al., 2019; Seurin et al., 2021; Guo et al., 2022), diversity-seeking (Bellemare et al., 2016; Hong et al., 2018; Seo et al., 2021; Yarats et al., 2021). These works propose various intrinsic motivation objectives which result in more efficient exploration than using random actions. A simple yet effective approach is Random Network Distillation (RND) (Burda et al., 2019) where the the predictions of a frozen network are used as distillation target for an online network, and the prediction error acts as an exploration reward. One significant drawback of intrinsic motivation methods is that they may need to cover large regions of sub-optimal state-action space before encountering extrinsic feedback. To mitigate this, approaches like Go-Explore (Ecoffet et al., 2019) reset to promising states and resume exploration from there, but this requires that the environment can be reset to desired states, or that arbitrary states can be reached via behavior (Ecoffet et al., 2021). Unfortunately, the ability to instantiate arbitrary states in the environment is not feasible in many domains of interest including Nethack, and particular configurations of the environment are not always reachable from all states.

**Learning from human demonstrations:** Equipping learning agents with priors derived from human demonstrations has long been an important direction of research. Behavior cloning has proven effective in complex tasks, for example, in the case of Minecraft (Shah et al., 2021; Baker et al., 2022; Fan et al., 2022). Video Pretraining (VPT, Baker et al., 2022) first trains an inverse dynamics model on a small dataset annotated by humans, and uses it to infer the actions in a large-scale unlabeled dataset. While drastically improving performance of RL agents on Minecraft, VPT does require an external source of supervision to provide labels. By contrast, incremental approaches first involve a reward-free or action-free pre-training phase, which can be used to gather missing actions or rewards, and then used to label the offline dataset or fine-tune the representations found during the first phase (Torabi et al., 2018a; Yu et al., 2021; Seo et al., 2022).

Another leading approach to learn from demonstrations is Generative Adversarial Imitation Learning (GAIL) (Ho & Ermon, 2016) where a discriminator model learns to distinguish expert and agent behavior, and the RL agent is incentivized to make its own distribution indistinguishable from the expert's. This was further extended to GAIfO (Torabi et al., 2018b) which learns to distinguish two consecutive observations instead of state-action distributions, for cases in which actions are not available. Similarly, BCO (Torabi et al., 2018a) extends behavior cloning to the action-free case, by learning an inverse model for action labeling from the agent's online data. ILPO (Edwards et al., 2019) extends BCO by factoring the action labeling problem into forward modelling conditioned on a latent action followed by remapping to the primitive action space. The learning process for ILPO involves making a prediction for each possible latent action in order to take the maximum likelihood prediction for estimating latent action labels, which can be computationally expensive for large state and action spaces, and is challenging when environment dynamics are stochastic. FORM (Jaegle et al., 2021) is another recently proposed approach that learns a forward generative model of dynamics from demonstrations without actions and rewards the agent for producing transitions that are likely under the model. Other approaches like R2D3 (Gülçehre et al., 2020) and DQfD (Hester et al., 2018) mix demonstrations and online data in the replay, but they require access to actions and rewards in the offline data, which is also the case in offline reinforcement learning (Fujimoto et al., 2019; Prudencio et al., 2022). As the demonstration data used in our experiments doesn't include any actions, we compare our approach with GAIfO, BCO, and FORM in this work.

**NetHack:** The game of NetHack (Küttler et al., 2020) is a complex pre-existing environment with a very sparse ultimate objective, large degrees of procedural generation and stochasticity, long-term dependencies, a strong notion of implicit monotonic progress, and a large state-action space. This makes *tabula rasa* learning on NetHack challenging, as recently emphasized by the results of the 2021 NetHack competition (Hambro et al., 2022a). Notably, symbolic and rule-based agents still outperform reinforcement learning methods on many metrics. In this work, we demonstrate that by leveraging openly available recorded gameplay from NetHack players in the wild (Hambro et al., 2022b), we can outperform state-of-the-art imitation and exploration approaches on a variety of hard NetHack tasks.

## 3 METHODOLOGY

Our method, which we refer to as Explore Like Experts (ELE), consists of two phases: i) learning a progress model offline, and ii) using the learned progress model as an auxiliary reward in reinforcement learning. The first phase of learning a progress model uses the expert data to learn a regressor which predicts a scalar progress reward given two observations from the same episode. The second phase simply uses the fixed weights of learned progress model to provide an auxiliary reward from the current observation and another past observation within this episode, and the underlying agent optimizes a weighted sum of the auxiliary and extrinsic reward while training.

Given a set of demonstrations $\mathcal{D}$, the goal of the progress model is to predict the temporal distance between two observations within the same demonstration. Formally, let $(s_i^d, s_j^d)$ be observations at the $i$th and $j$th position in the demonstration trajectory $T_d$. We predict $f(s_i^d, s_j^d; \theta)$ denoting the temporal distance between $s_i^d$ and $s_j^d$ in the trajectory. Expert demonstrations can have very long episodes, so we predict the distance between these observations in signed log-space: $y_{ij}^d = \text{sgn}(j-i)\log(1+|j-i|)$. The progress model is trained by minimizing the mean squared error between the true value $y_{ij}^d$ in transformed space and $f(s_i^d, s_j^d; \theta)$. The resulting progress model $f(.; \theta)$ captures a monotonic increasing function of progress within the expert trajectories. In NetHack, this corresponds to behaviors such as increasing the character's level, revealing tiles in the dungeon, discovering new dungeon levels, finding gold, and improving the hero's armor. See Fig. 5 and Section 4.5 for visualizations of the progress model on example trajectories.

The second phase uses the learned progress model $f(;\theta^*)$ with a reinforcement learning algorithm $A$. In this phase, the extrinsic reward of the environment is augmented with the auxiliary reward from the progress model. To calculate the reward, we transform the progress estimates back to linear space: $f^*(s_i,s_j;\theta^*)=\text{sgn}(f(s_i,s_j;\theta))*(e^{f(s_i,s_j;\theta^*)}+1)$. While the auxiliary reward could be computed in many ways using the progress model, such as computing the temporal distance between the initial state and the current state, or computing the relative progress against the initial observation for consecutive states, we empirically observe that computing progress over small distances is significantly more effective (see Section A.4 for performance of the different variants). Hence in this work, we compute an auxiliary reward $r_t^a=f^*(s_{t-k},s_t;\theta^*)$ to measure progress between the current state $(s_t)$ and a state from $k$ steps in the past $(s_{t-k})$ where $k$ is a hyperparameter. The reward $r_t^{\text{total}}$ is computed as weighted sum of auxiliary reward $r_t^a$ and extrinsic environment reward $r_t$:

$$r_t^{\text{total}} \leftarrow r_t + w^a f^*(s_{t-k},s_t;\theta^*) \tag{1}$$

where $w^a>0$ is a hyperparameter. See Algorithm 1 for a pseudocode description.

## 3.1 GRID WORLD EXAMPLE

To illustrate the progress model in a simple case, we consider a tabular grid-world environment with a Markov decision process as follows: a 2D state space $s \in [1,200] \times [1,200]$, where at each timestep the agent can move in any of the eight cardinal and diagonal directions. The agent begins the episode at $s_0 = (1,100)$ and receives a reward of 1000 when it reaches $s = (200,100)$, and zero otherwise. The reward and termination conditions are designed to be similar to the NetHack tasks that we investigate in Section 4. In this toy example we employ a tabular Q-learning approach, with the the vanilla RL agent only receiving extrinsic reward, while our ELE agent additionally receives exploration reward from the progress model. The progress model in this case is also tabular with an input space of $s_i \times s_j$. It is trained to estimate progress on a set of demonstrations from a sub-optimal $\epsilon$-greedy oracle policy, where an action is chosen uniformly at random with probability $\epsilon=0.975$ and the optimal action is chosen with probability $1-\epsilon=0.025$. In other words, the demonstration agent performs a biased random walk toward the goal.

We demonstrate in Fig. 2 that the paths taken by the tabular Q-learning agent without an exploration reward do not approach the shortest path even after 20k episodes, whereas the same agent with the addition of the ELE objective achieves much more direct paths despite learning from highly sub-optimal demonstrations. As shown in Fig. 2, even trained only on extremely noisy demonstrations, the progress model forms a smooth and consistent flow field toward the goal, due to the non-local nature of our approach. In contrast, most existing imitation learning approaches are purely local, and their models incorporate information only in the near vicinity of the current state. See Section A.8 for more details on the tabular case and a theoretical analysis in terms of stochastic processes.

In the rest of the paper, we take NetHack as the domain of interest. It is also grid-based, but is a much more complex environment than the toy example, is a pre-existing domain that humans have found engaging to play for over 30 years, and is still far from solved by any existing approach (Küttler et al., 2020; Hambro et al., 2022a). The method is not specific to NetHack and applies in principle to other domains that admit a notion of long-term progress, where demonstration data is available.

## 4 EXPERIMENTS

In this work, we consider NetHack as the primary environment for evaluating our proposed approach. ELE and all baselines are implemented as auxiliary rewards and/or losses on top of an underlying reinforcement learning agent. We use Muesli (Hessel et al., 2021a) as the agent, which is a lightweight modification of MuZero (Schrittwieser et al., 2020) that does not involve deep tree search. In these experiments we use a progress reward scale $w^a=0.01$ and progress history length $k=8$; see Section A.1 for more details about the implementation. All reported results are aggregated over 5 seeds. The progress model takes the form of a neural network with the same structure as the feed-forward component of the underlying agent, but weights are not shared with the agent. See A.1 for implementation details about the architecture.

For all NetHack experiments, we remove the explicit timer from the observations, as it provides a shortcut to predicting timing differences that does not reflect the important elements of progress in the game (a common practice in exploration-focused work (Bellemare et al., 2016)). See Section A.6 for more information on timer sensitivity. All methods were implemented using the Jax ecosystem for scalable differentiable

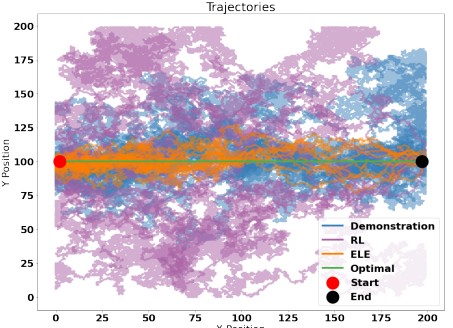
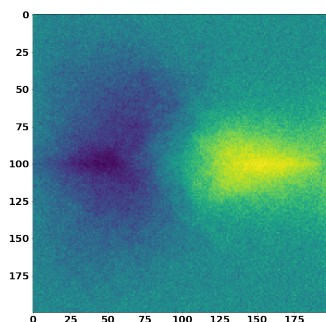

Figure 2: Toy grid environment: $200 \times 200$ grid with tabular Q-learning agent and ELE. The agent can move in 8 directions, starts at $(1, 100)$ and receives a reward of 1000 when reaching $(200, 100)$. See Section A.8 for more details. Left: 10 example trajectories from each method: ELE discovers paths that reach the goal more directly than either pure RL with $\epsilon$-greedy random exploration or the demonstrations that it learned from. Right: Visualization of the ELE progress model $f^*(s_i, s_j)$ for $s_i = (100, 100)$ against all possible successor states $s_j$. The model has learned that expert progress tends to be positive (bright) to the right of $s_i$ and negative (dark) to the left.

programming (Babuschkin et al., 2020; Hessel et al., 2021b; Yang et al., 2021). We use demonstrations from the publically available expert trajectories recorded at http://nethack.alt.org.

## 4.1 TASKS

We evaluate our approach on three standard tasks: **Score**, **Scout**, and **Oracle** (Küttler et al., 2020), as well as four new sparse tasks that represent important subtasks in the full NetHack game. Importantly, for all of these tasks we do not use the reduced action space presented in (Küttler et al., 2020), but instead use the entire unrestricted NetHack action space of 121 discrete actions, and we do not skip any messages or menu states. In addition, the hero role is chosen uniformly at random from the entire pool every episode, which is a challenging setting even for humans, who often exhibit a preference for particular roles. These settings ensure that the tasks that we investigate represent the entire complexity of the early game. For every task, episodes are terminated after 1M environment steps or when the agent dies, except as described below. All agents are all trained for a total of 1B environment steps, the standard frame budget established in the literature (Küttler et al., 2020). To convey a sense of the relative time horizon of each task, we list the average episode lengths as achieved by our method in Table 1.

In the **Score** task, the reward function is the difference in the in-game score from the previous timestep, as is commonly done in game-based environments (Bellemare et al., 2013). This is a relatively dense reward signal: score in NetHack is obtained for killing monsters, obtaining gold, identifying wands and potions, descending into deeper dungeon levels, reading novels, purchasing consultations from the oracle, and more.

In the **Scout** task, the agent is given a reward of 1 for every tile revealed in the dungeon. This is the densest task that we consider in this work, and provides an explicit signal for learning spatial exploration behavior.

The objective of the **Oracle** task is to find the Oracle in the dungeon, which is a non-player character that is always located in a distinct location between 4 and 8 dungeon levels below the starting level. Once the agent moves to a tile adjacent to the Oracle, it receives a reward of 1000 and the episode is terminated. This is a very sparse and fairly challenging task: in our human baselines, we find that intermediate human players are only able to reach the Oracle in approximately 48% of episodes.

In addition to these standard tasks, we study four more tasks of increasing sparsity to provide a continuum of challenge between the **Score** and **Oracle** tasks. For the **Depth 2** and **Depth 4** tasks, the objective is to reach a dungeon level 2 or 4 levels below the surface. The initial level is at depth 1, so these tasks require navigating 1 and 3 dungeon levels to find the stairs, respectively. As with **Oracle**, the agent is given a reward of 1000 when reaching the target depth, and the episode is terminated. The **Exp Level N** tasks reward the agent for reaching a target experience level. The hero begins the game at level 1, so **Exp Level 2** and **Exp Level 4** require defeating enough monsters to gain 1 and 3 levels, respectively. As with the previous sparse tasks, the agent receives a reward of 1000 once the target experience level is reached, and the episode is terminated.

In order to facilitate comparison between tasks, we normalize episode returns to lie approximately in the range [0,1]. In the sparse tasks, we simply scale the reward down from 1000 to 1. For **Score** and **Scout**, we divide all reported returns by the 80th percentile of the episode returns of all methods (1410 and 3425, respectively). Note that this is for reporting purposes only: the agents optimize the true underlying task objective.

| Task | Length | | Task | Length | | Task | Length |
|---|---|---|---|---|---|---|---|
| **[Std]** Scout | 596.2 | | Depth 2 | 76.1 | | Depth 4 | 183.7 |
| **[Std]** Score | 511.5 | | Exp Level 2 | 133.4 | | Exp Level 4 | 386.6 |
| **[Std]** Oracle | 547.2 | | | | | | |

Table 1: Average episode length for each task as achieved by ELE; tasks marked with **[Std]** indicate standard tasks that were defined in (Küttler et al., 2020). Since the action space of NetHack consists of 121 different actions, sparse tasks with horizons as long as these are very challenging to solve without a dedicated exploration strategy, and many are still challenging even with access to human demonstrations (see Section 4.4). Unlike real-time games with high frame rates, the turn-based nature of NetHack means that even relatively short episodes such as those in the **Depth 2** task can correspond to multiple minutes of human gameplay, and the **Oracle** task can take intermediate human players more than 30 minutes of real time to complete (see Section A.3 for details).

## 4.2 DATASETS

We leverage the data recorded and made publically available by the owners of http://nethack.alt.org (Hambro et al., 2022b) in order to provide demonstrations for learning to play NetHack. The data poses at least three distinct challenges: 1) the recordings do not include actions or rewards, so straightforward behavior cloning or GAIL cannot be applied; 2) users play with a wide variety of interface settings, including but not limited to character and color remapping, user interface element repositioning, and auto-pickup and other game settings (Hambro et al., 2022b); and 3) the gameplay varies widely in quality: many beginners use the server as their primary platform for playing the game, and the data also includes scripted bots that do not play at high levels of competence or at all (e.g. behaviors such as save-scumming (Hambro et al., 2022b)). However, as the server is the most popular platform for playing the game, it also contains large amounts of highly skilful play. As the data was recorded over the last 14 years, the players were not aiming to solve the tasks we consider in this work; however, the tasks represent common milestones encountered in natural gameplay.

Our method addresses 1) and 2) by not relying on actions or a sensitive discriminator to function. 3) remains an issue, since large proportions of deliberately bad gameplay are likely to impact any learning algorithm that attempts to leverage demonstrations to improve performance. To mitigate this, we use data from only the top 10 users ranked by a quantity called *Z-score* (nethackwiki.com, 2022) that rewards decreasing values for each win, which is calculated as $Z_{user} = \sum_{role} \sum_{i=1}^{M_{role}} \frac{1}{i}$ where $M_{role}$ is the total number of games the user has won with that role. The Z-score metric has been designed by the community to incentivize winning the game with a wide selection of character roles, which aligns with our goal for a dataset to contain as much high-quality gameplay across the entire space of the game as possible.

To empower further research using high-quality NetHack demonstrations, we have released a curated dataset of gameplay from the top 10 users in the form of easily accessible compressed data at https://github.com/deepmind/nao_top10. In total, this amounts to 16478 sessions of gameplay with a total of approximately 184M transitions, and is approximately 12 GB in size. See Section A.2 for more information on the datasets. The resulting dataset is much smaller than the entire dataset in its original form.

## 4.3 BASELINES

We compare our method against a variety of popular baselines that are focused on exploration and learning from demonstrations without access to actions, as well as the underlying Muesli agent without any explicit exploration objectives. All baselines use the same curated demonstration set that we use for ELE. We performed a hyperparameter search for each baseline; see Section A.1 for implementation details and hyperparameter search results.

**FORM** (Jaegle et al., 2021): an action-free imitation learning approach that learns a passive dynamics model $p_D(s_{t+1}|s_t)$ from expert demonstrations, and uses the log-likelihood ratio of the expert dynamics model to an online estimator of its own transition dynamics $\log p_D(s_{t+1}|s_t) - \log p_I(s_{t+1}|s_t)$ as an imitation reward.

**BCO** (Torabi et al., 2018a): an approach to behavior cloning without access to actions in which an inverse dynamics model $p(a_t|s_t,s_{t+1})$ is trained on the agent's own transitions, and used to label the action-free expert data to enable a behavior cloning objective. It should be noted that although the demonstrations are not recorded for tasks considered in this work, the BCO loss acts as a policy regularizer toward the types of behavior that are generally useful for the game. We also add the extrinsic reward as an objective to all baselines: the purpose of the imitation losses is to encourage exploration.

**GAIfO** (Torabi et al., 2018b): an adversarial approach to action-free imitation learning in which a discriminator $p(\text{Expert}|s_t,s_{t+1})$ is trained to distinguish agent and expert transitions, and the probability of the agent's transitions being classified as expert behavior is used as an imitation reward.

**RND** (Burda et al., 2019): a demonstration-free exploration reward in which a randomly initialized feature encoder $f_{\text{frozen}}(s_t)$ is used as a target for distillation into an online trained encoder $f_{\text{online}}(s_t)$, and the prediction error between the online and target encoders is used as an intrinsic reward.

## 4.4 RESULTS

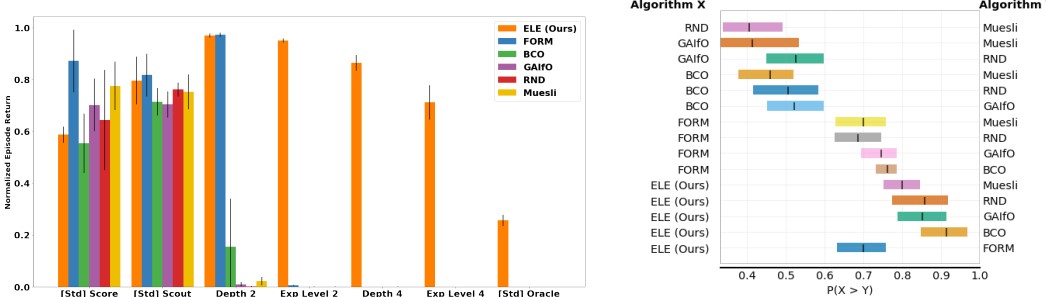

Figure 3: Left: Median normalized episode return achieved by each approach after 1B frames of training, with error bars denoting the range from the 25th to 75th percentile over 5 seeds. Tasks marked as **[Std]** are existing tasks from the literature. The sparse task reward is scaled down by 1000 to lie in the range $[0,1]$; for **Score** and **Scout** we normalize the return by the 80th percentile over all approaches (1410 and 3425, respectively) for ease of comparison. Right: Permutation test comparing all pairs of algorithms on randomly sampled runs, to quantify the probability that each algorithm outperforms the others on any given task (Agarwal et al., 2021). ELE outperforms all approaches with a wide margin, except FORM which we outperform in 70% of samples.

We evaluate our approach on the tasks in Section 4.1. Fig. 3 (left) shows the total episode return achieved by each approach on all of the tasks we investigate after training for 1B steps, which is an established frame budget for NetHack (Küttler et al., 2020). Figure 3 (right) shows the permutation test comparing all pair of algorithms and statistically evaluating if one algorithm is better than other as proposed in Agarwal et al. (2021). See Section A.5 for further comparison on aggregate metrics based on the mean, median, IQM and optimality gap.

The baseline approaches do well on the denser tasks, but struggle to discover the extrinsic reward on the sparse tasks. By contrast, ELE succeeds even on the very sparse tasks, and achieves similar results to the other approaches on the dense tasks. Note that the **Oracle** task is challenging even for human players: in our human baseline gameplay, intermediate human players achieve approximately 48% success on average. See Sections 4.2 and A.6 for more details about the demonstrations and further analysis.

## 4.5 ANALYSIS

In Fig. 4, we plot the (left) cumulative local progress and (right) instantaneous progress with respect to the initial state, for 100 different episodes from a variety of different data sources. As expected, progress increases consistently for all datasets except for the uniform random policy. These plots demonstrate that

agents are able to optimize progress more aggressively than either the source data or our human baseline recordings, which we also observe qualitatively in the relatively reckless behavior of the agents—humans tend to be much more careful and conservative. The high variance of the instantaneous progress plot shows the difficulty of estimating temporal differences for long timescales, and justifies the use of cumulative local progress as an exploration objective instead.

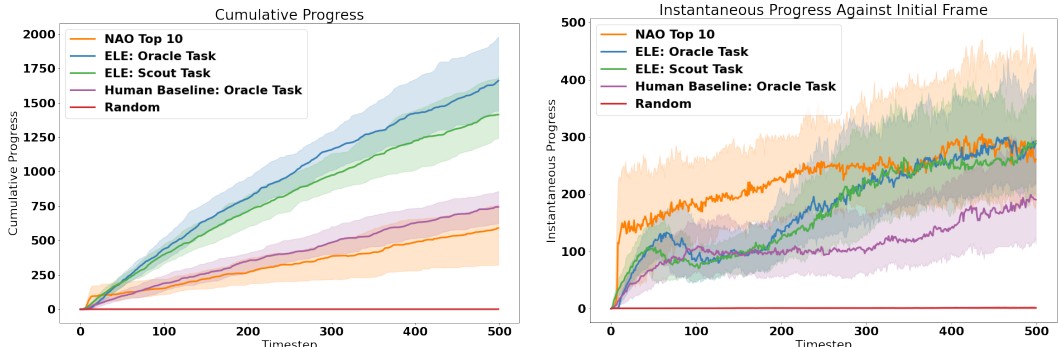

Figure 4: Example progress curves on the first 500 steps of recorded episodes. Each curve shows the median and middle 50% of the data over 100 episodes. Left) Cumulative progress: the sum of $f^*(s_{t-8}, s_t)$ over the episode. The agents pursue progress much more aggressively than either the NAO experts or our on-task human baseline. Right) Instantaneous progress against the initial frame: $f^*(s_0, s_t)$. Instantaneous progress estimates against the initial frame are much noisier than the cumulative sum of local progress, in part because long-range progress estimates are much more difficult due to the many possible paths between distant states.

Fig. 5 shows the instantaneous progress estimates $f^*(s_{t-k}, s_t)$ for a single representative episode, with the peak of maximum progress indicated in red, alongside a rendering of the two frames for the transition corresponding to this peak, and an example saliency map of the progress model on the bottom two statistics lines in the observation. Fig. 5 (bottom) shows the sensitivity of the progress model to features including character and dungeon level, score, gold, and armor class. In other words, ELE has learned that revealing tiles in the dungeon, increasing its ability scores, and descending to deeper levels all reflect expert progress. Despite the procedural generation and stochasticity of the underlying environment, some aspects of the observations reliably indicate progress in the game. These measures indicate that ELE's progress model captures many of these subtle indicators from even such a diverse set of demonstrations.

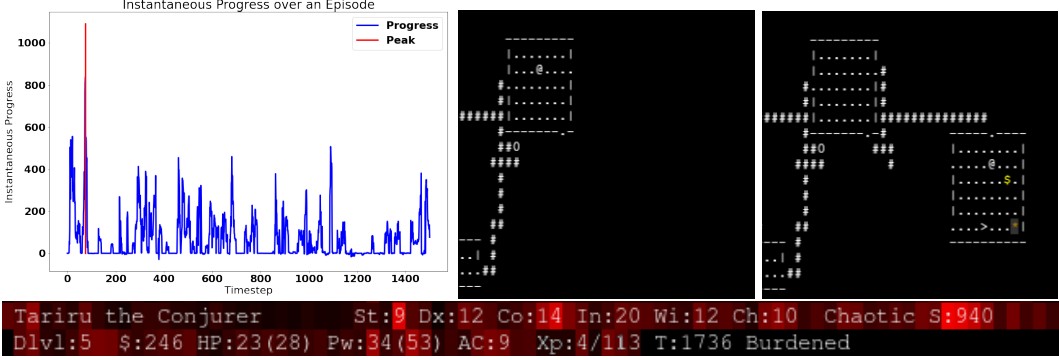

Figure 5: Left: Instantaneous progress $f^*(s_{t-k}, s_t)$ for a representative episode where $k = 8$, with peak progress indicated in red. Right: Cropped frames $s_{t-k}$ and $s_t$ corresponding to the peak progress estimate. This transition demonstrates how dungeon exploration results in high progress estimates. Bottom: Example saliency map over the bottom two lines of the observation, with brighter colors corresponding to higher sensitivity of the output to that cell in the observation. The progress model has learned to be sensitive to dungeon and experience levels, character ability scores, gold, magic power, armor class, and game score. See Fig. 17 for more saliency visualizations.

Figure 6 shows the results of experiments using different history offsets and dataset sizes for ELE. On the left, the learned progress model is evaluated using different offsets $k$ into the recent past for the progress reward on **Oracle**, the most challenging task. While ELE is robust to a variety of history lengths, we observe that horizon length $k=8$ performs best for sparse tasks. On the right, we evaluate the performance of ELE with increasing dataset size by using demonstrations from the top N experts by Z-score. While large datasets perform well on the dense **Score** task, we find that the top 10 experts provide the best dataset for the challenging **Oracle** task. In Section A.2 we provide an analysis of the datasets and demonstrate that Top 10 provides the best balance between dataset size and coverage of the early game, which is where the tasks we investigate in this paper take place.

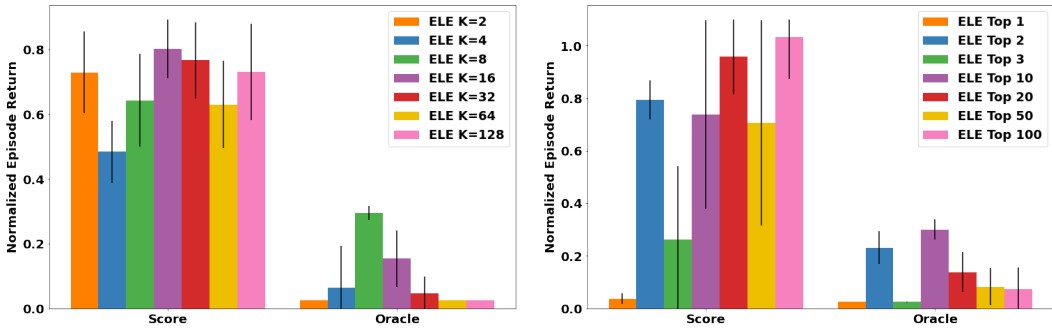

Figure 6: Comparison of ELE using different hyperparameters. Left: Hyperparameter sweep for history length $k$ where $k=8$ performs best. Right: Performance of ELE using demonstration datasets of increasing size. We find that the top 10 experts provide the best dataset for our most challenging task (see Section A.2 for further analysis).

## 5 CONCLUSION

In this work, we hypothesize that a collection of expert demonstrations implicitly defines a set of monotonic progress functions, and that increasing these progress functions over time results in efficient exploration in challenging environments. We propose a method for approximating one simple class of such monotonic progress functions: the temporal difference between frames. All else being equal, exploration that looks like positive temporal difference under the expert distribution is likely to discover extrinsic task feedback more efficiently than unguided exploration. We demonstrate in the NetHack environment that following a progress model is an effective way to discover rewards in even very sparse tasks, and even when the demonstrations in question consist of noisy, action-free data captured in the wild.

When the observations contain explicit timing information, effort is required to mitigate the tendency of the progress model to focus solely on these features to the exclusion of more semantically meaningful information. In this work we explicitly remove the timer from the input, which is easy to do in NetHack observations. However, for other environments where the explicit timing information is more subtle and difficult to remove explicitly, there are promising research directions involving unsupervised approaches to factor these features out.

The issue with explicit time observations is an instance of a more general challenge in applying this sort of method. Consider tasks with cyclic dynamics, or state maintenance tasks such as balancing a pole: maximizing predicted temporal distances would not be sufficient, and in this situation an explicit timer in the observation may actually help rather than hurt. Thus, when applying a method such as ELE, some care is warranted to ensure that the observations will encourage the model to learn a notion of progress that is appropriate to the task.

The temporal difference between frames is a simple monotonic function that we have chosen to focus on in this work. However, we hypothesize that demonstrations implicitly define a large set of such functions, and discovering latent progress functions not directly tied to temporal differences is another important area that deserves further investigation beyond the scope of this report.

ACKNOWLEDGEMENTS

We would like to thank Duncan Williams, Surya Bhupatiraju, Dan Horgan and Richard Powell for developing the underlying Muesli agent for NetHack, including hyperparameter and architecture tuning. We would like to thank Adam Santoro, Doina Precup and Tim Rocktäschel for feedback on the paper, which improved the manuscript substantially. We would also like to thank the anonymous reviewers for detailed reviews and feedback on the paper. Additionally, we are grateful to M. Drew Streib, the host of the nethack.alt.org server for recording 14 years of gameplay, without which this work would not have been possible. Last but not the least, we would like to extend our heartfelt thanks to the 10 experts in the NetHack community whose gameplay data helped us to make scientific advances in this domain, in order of increasing Z-score: stenno, 78291, Fek, YumYum, rschaff, oh6, Luxidream, Tariru, stth, and Stroller.

ETHICS STATEMENT

This work is made possible by expert data generated by the NetHack community and made publically available by the owners of `http://nethack.alt.org`. The data includes full human gameplay in the form of terminal recordings, and these recordings include the usernames of the humans that generated the data. This data is already publically available and permissively licensed; we train on the existing data and release a curated subset.

REPRODUCIBILITY STATEMENT

We have made an effort to ensure reproducibility via a) releasing the dataset that we used for our main results to empower the community to conduct further research using this curated subset of high-quality play, and b) exhaustively detailing the hyperparameters and implementation details of our approach, our network architecture, and the computational resources used for our experiments in Section A.1. We also detail several potential variations of our approach in Section A.4.

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

# A  APPENDIX

## A.1  IMPLEMENTATION DETAILS

Architecture diagrams for all approaches are shown in Fig. 7. The architecture of the agent is composed of a feedforward ResNet (He et al., 2016) torso without normalization layers, followed by an LSTM (Hochreiter & Schmidhuber, 1997) and a residual connection from the torso. The BCO baseline uses the same architecture as the agent. The GAIfO and RND baselines use only the feedforward network, with each hidden layer reduced to 16 channels to minimize overfitting. FORM requires full state predictions, and so we use the fully-convolutional architecture shown in Fig. 7.

Hyperparameters for all experiments are shown in Table 2. Where hyperparameters differ between approaches, the differences are shown in Table 3. We anneal the learning rate linearly toward its final value over the course of 1B frames. Each experiment was run on 8 TPUv3 accelerators using a podracer configuration (Hessel et al., 2021b). Algorithm 1 provides a pseudocode description of the algorithm. Results of hyperparameter sweeps for each approach are shown in Fig. 8.

| Hyperparameter | Value |
|---|---|
| Optimizer | Adam (Kingma & Ba, 2014) |
| Adam $\epsilon$ | $10^{-8}$ |
| Initial learning rate | $6\times10^{-4}$ |
| Final learning rate (linear anneal) | $10^{-5}$ |
| Maximum absolute parameter update | 1.0 |
| Discount factor | 0.997 |
| Policy unroll length | 35 |
| Model unroll length | 5 |
| Retrace $\lambda$ (Munos et al., 2016) | 0.95 |
| Batch size | 960 sequences |
| Replay proportion in batch | 50% |
| Replay buffer capacity | 4000 frames |
| Replay max samples per insert | 1 |
| Target network update rate | 0.1 |
| Muesli regularization weight (uniform prior) | 0.003 |
| Muesli regularization weight (target network prior) | 0.03 |
| Value loss weight | 0.25 |
| Reward loss weight | 1.0 |
| Policy loss weight | 3.0 |

Table 2: Muesli hyperparameters that are consistent between methods.

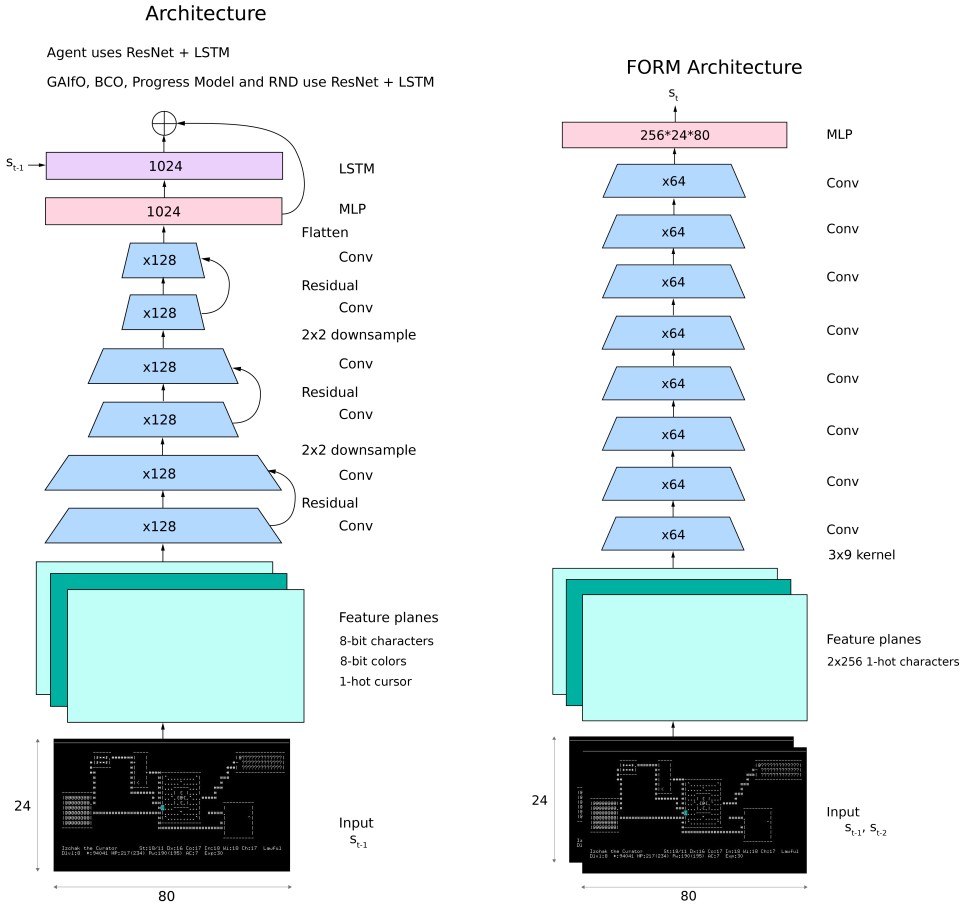

Figure 7: Left: RL agent architecture. Right: FORM model architecture.

| Hyperparameter | Value |
|---|---|
| ELE reward weight $w^a$ | 0.01 |
| BCO loss weight | 0.01 |
| FORM batch size | 576 sequences |
| FORM reward weight | 0.1 |
| GAIfO reward weight | 1.0 |
| RND reward weight | 0.1 |

Table 3: Hyperparameters that differ between methods. Note that the large prediction network in FORM requires a smaller batch size in order to fit in accelerator memory.

## A.2 DATASET SIZE

The full dataset from `http://nethack.alt.org` (Hambro et al., 2022b) contains approximately 7M sequences at the time of this writing. However, the data varies widely in quality from beginners and scripted bots all the way to world-class gameplay from the top NetHack experts. In addition, the full dataset is too large to fit comfortably on most storage hardware. As a result, it would be desirable to reduce the size of the dataset to just a curated sample of the highest quality play. To generate a curated subset of the highest quality data, we filter the dataset to include only the sessions from the top N users as ranked by *Z-score*. From the NetHack wiki (nethackwiki.com, 2022):

> Z-score is meant to provide an alternative to using score to rank players, since score
> is not a reliable measure of ability. It is compiled by looking at all of a player's

---

**Algorithm 1:** Pseudocode description of Explore Like Experts (ELE)

**Input** : Dataset $\mathcal{D}$, hyperparameters $k$ and $w^a$

1 **for** *epoch* $m=1,2,..,M$ **do**
2      Sample minibatch $\{(s_i^d, s_j^d)\} \sim \mathcal{D}$
     /* Update the parameters of the progress model            */
3      $\theta_{m+1} \leftarrow \theta_m + \alpha * \Sigma_{i,j,k} \nabla (y_{ij}^d - f(s_i, s_j; \theta_m))^2$
4 **for** *epoch* $j=1,2,...J$ **do**
5      Sample a trajectory $\tau \in \mathcal{T}$ from current actor policy
6      **for** $(s_t, a_t, r_t) \in \tau$ **do**
         /* Compute auxiliary reward using the progress model       */
7          $r_t^a = f^*(s_{t-k}, s_t; \theta_M)$
         /* Update the total reward by linear
           combination of environment reward and auxiliary reward    */
8          $r_t \leftarrow r_t + w^a r_t^a$
9      Update agent parameters with $\tau$ using new reward

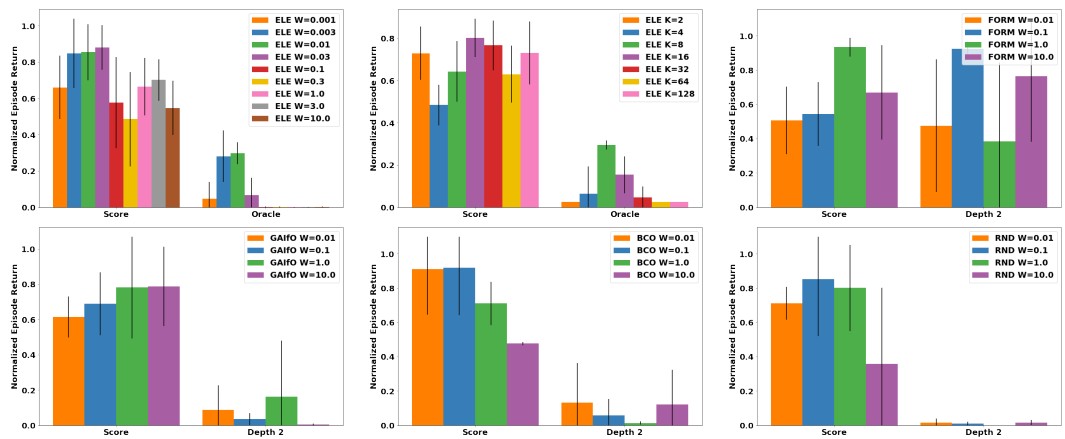

Figure 8: Hyperparameter sweeps for each method on a dense and a sparse task. Where there is ambiguity over which hyperparameter value to use for the final experiments, we choose the hyperparameter that performs best on the sparse task over the dense task.

ascensions and assigning 1 point for the first ascension of a role, 1/2 a point for the second ascension, 1/3 for the third, and so on. Z-score encourages playing the complete set of roles rather than repeatedly ascending the same role.

In addition to filtering by Z-score, we include only episodes where the screen contains only standard ASCII characters (i.e. sessions that use the standard NetHack interface), we eliminate any episodes where we cannot identify the dungeon grid on the screen (as this indicates the user may be using a non-standard user interface layout), and consider only episodes that contain at least 100 timesteps to minimize the number of episodes that involve save-scumming (Hambro et al., 2022b).

In order to determine how many of the top experts are sufficient to use as a demonstration dataset we evaluate our method on **Score**, the most popular NetHack task, and **Oracle**, the most challenging of the sparse tasks in this paper, using demonstrations from the top N users for $N \in [1,2,3,5,10,20,50,100]$; the results are shown in Fig. 6 (right). Dataset sizes are shown in Table 4. All else being equal, one might expect larger datasets to be better. In actuality, we find that the top 10 experts works best for the sparse tasks, and so this is the dataset we use for all of the experiments in this work that use demonstrations, and this is the dataset we release alongside the paper.

In order to explain why increasing dataset size does not yield better results on these tasks, we consider two analyses. Fig. 9 shows, for each dataset, the proportion of the timesteps in the dataset that correspond to the early game of NetHack, i.e. the portion of the game that is most relevant to all of the tasks we

| Dataset | # Sequences | # Timesteps |
|---------|-------------|-------------|
| Top 1   | 4703        | 54.2 M      |
| Top 2   | 8336        | 112.1 M     |
| Top 3   | 11887       | 136.3 M     |
| Top 5   | 11962       | 136.5 M     |
| Top 10  | 16478       | 184.2 M     |
| Top 20  | 33882       | 282.2 M     |
| Top 50  | 62300       | 421.3 M     |
| Top 100 | 98614       | 567.5 M     |

Table 4: Size of each dataset considered in this section.

consider. Specifically, this is experience levels 1-5, dungeon levels 1-10, and dungeon number 0 (i.e. the main dungeon). We see in these results that the top-10 dataset which we find yields the best results is also the dataset that includes the most gameplay covering the early dungeon and experience levels.

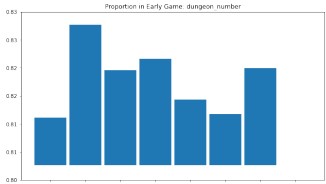 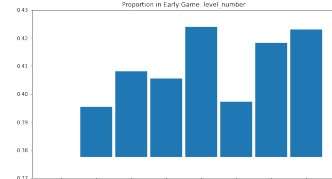 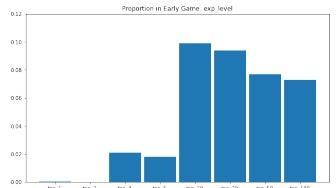

Figure 9: Proportion of transitions that cover the early game in each dataset; higher is better. The top-10 dataset contains the largest proportion of early game data in terms of both dungeon level and experience level, and is only marginally behind top-2 in terms of time spent in the main branch of the dungeon (dungeon number zero).

In order to attempt to quantify the degree to which each dataset covers the state distribution of the game, we also analyze the difference between each dataset and all of the others as measured by the total symmetric KL divergence between distributions over: character alignment, role, dungeon number, dungeon level, and experience level (See Fig. 10). We see with this analysis that the top-10 dataset is closest to all of the other datasets by most of the quantities in question, which suggests that top-10 may have the best general coverage of the game.

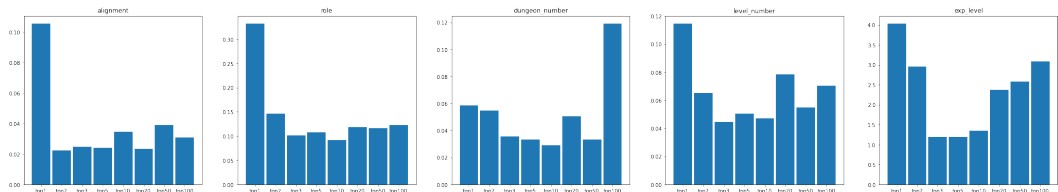

Figure 10: Total KL between distributions over alignment, role, dungeon number, level number, and experience level, as the sum over the KL between the dataset in question and all others. Lower values indicate that the dataset coverage is closer to the others.

### A.3 HUMAN BASELINE FOR ORACLE TASK

To convey a sense of the difficulty of the **Oracle** task, we collected 170 episodes of gameplay from an intermediate NetHack player. Table 5 reports some statistics of these human games. In particular, our human baseline finds the Oracle in 48.2% of episodes, taking an average of 899.2 timesteps to do so on a successful episode, which corresponds to an average of 8.9 minutes of gameplay. This is longer than a typical episode from our agent, which completes the task in an average of 547.2 steps as reported in Table 1. On average the human player performed 1.9 steps per second, which highlights how much longer a game of NetHack can be, compared to environments with high frequency real-time frame rates such as Atari.

| | |
|---|---|
| Number of games | 170 |
| Success rate | 48.2% |
| Number of timesteps (All) | $863.5 \pm 584.8$ steps |
| Number of timesteps (Successful) | $899.2 \pm 662.9$ steps |
| Maximum number of timesteps | 4663 steps |
| Real time length (All) | $8.2 \pm 5.9$ minutes |
| Real time length (Successful) | $8.9 \pm 7.0$ minutes |
| Maximum real time length | 35.1 minutes |
| Steps per second | $1.9 \pm 0.5$ |

Table 5: Statistics of intermediate human gameplay on the challenging **Oracle** task.

### A.4 PROGRESS REWARD VARIANTS

Given a progress model $f^*(s_i,s_j)$ that estimates the temporal distance between two states, there are a variety of ways that an exploration reward can be formulated. In this section, we explore three potential approaches. With the **Initial Absolute** reward function, we give an exploration reward to the agent equal to the estimated temporal distance between the current state and the initial state $r_t = f^*(s_0,s_t)$. The **Local Absolute** reward function corresponds to a reward function equal to the estimated temporal distance between the current state and a frame $k$ steps in the past $r_t = f^*(s_{t-k},s_t)$. Finally, the **Local Relative** approach rewards the agent based on the difference in local progress between adjacent states $r_t = f^*(s_{t-k},s_t) - f^*(s_{t-k},s_{t-1})$.

Fig. 11 shows the performance of the agent using each of the variants on **Depth 4** and **Oracle**, two of the more challenging sparse tasks. We can see that the local absolute progress performs best, with local relative progress achieving reasonable performance, but worse than local absolute. Progress against the initial frame, on the other hand, performs quite poorly—see Fig. 4 for a visualization of the initial absolute progress over the course of an episode: it has very high variance, indicating that it is difficult to make absolute progress predictions over such long timescales.

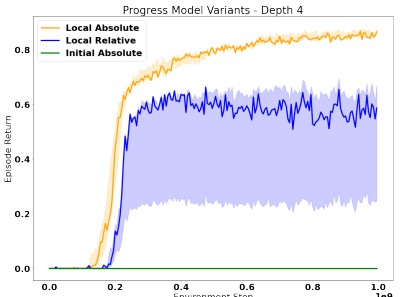 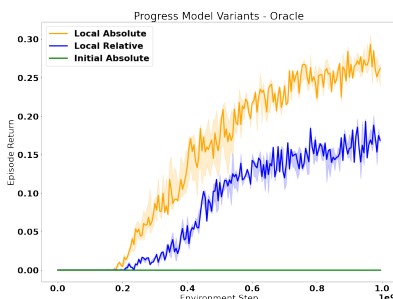

Figure 11: Performance of three variants of the progress reward: **Local Absolute** refers to the approach that we use in the paper where the reward $r_t = f^*(s_{t-k},s_t)$ with $k=8$; **Local Relative** corresponds to a reward based on the difference between local progress for adjacent states $r_t = f^*(s_{t-k},s_t) - f^*(s_{t-k},s_{t-1})$; and **Initial Absolute** refers to rewarding progress against the initial state of the episode $r_t = f^*(s_0,s_t)$. This figure shows the performance of the three variants on two of the challenging sparse tasks: **Depth 4** (Left) and **Oracle** (Right).

## A.5 STATISTICAL ANALYSIS OF RESULTS

In this section we conduct a statistical analysis of the performance of our approach using the methouds outlined in (Agarwal et al., 2021). Fig. 12 shows evaluation metrics aggregated across all tasks in order to produce a larger statistical sample. As mentioned in Section 4.1, we normalize the returns of the dense tasks **Score** and **Scout** by the 80th percentile across all approaches so they lie approximately in a comparable range $[0,1]$ to the sparse tasks. See Fig. 3 (right) for permutation tests that demonstrate that our method outperforms RND, GAIfO, BCO, and vanilla Muesli at confidence levels of at least 85% on this benchmark, while since FORM performs well on **Depth 2** as well as the dense tasks, ELE only outperforms FORM with confidence of approximately 70%.

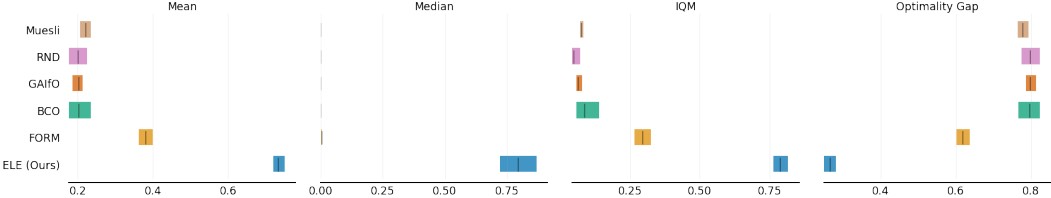

Figure 12: Aggregate metrics on 9 NetHack tasks with 95% CIs based on 5 random seeds. Higher mean, median and IQM scores and lower optimality gap are better. Our method ELE significantly outperforms the baselines according to all measures of central tendency.

## A.6 ANALYSIS OF THE PROGRESS MODEL

In order to visualize the sort of transitions that result in large progress events, we include a representative example of an instantaneous progress curve $f^*(s_{t-8}, s_t)$ for a typical episode generated by our ELE agent. Fig. 13 shows the progress values with the top 4 progress events highlighted in red, and Fig. 14 shows the specific transitions that correspond to those events. The progress model has learned that, in this episode, revealing tiles in the dungeon, leveling up, and descending to new dungeon levels are the most prominent indicators of progress.

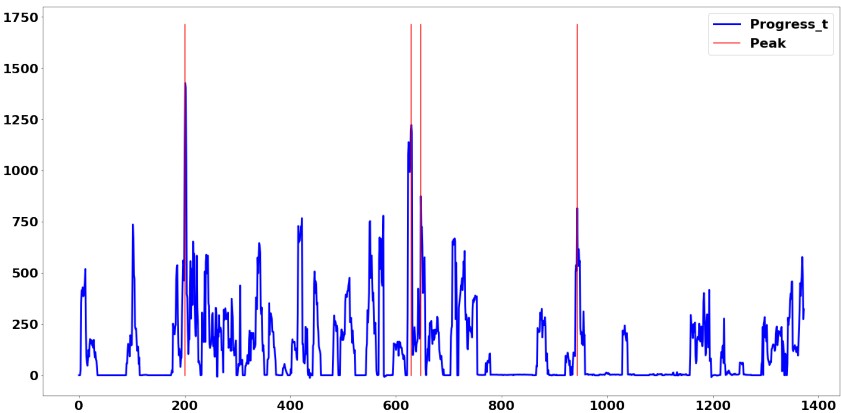

Figure 13: Peaks of the progress estimate during a representative episode.

In the latest version of the NetHack learning environment, the in-game timer is visible in the observations. This is an issue for our approach, where if trivial timing information is available, the progress model is not required to learn semantically meaningful representations of progress. Therefore, for all methods evaluated in this work, we mask the timer feature with zeros to reduce reliance on this trivial information. See Fig. 15 for a) the loss curve of the supervised regressor in the offline training phase for the model with and without access to the timer feature, and b) the sensitivity of the model to the timer feature as a proportion of the total gradient with respect to the input. In Fig. 16 we show the performance of the resulting agent when trained against a progress model with and without access to the timer.

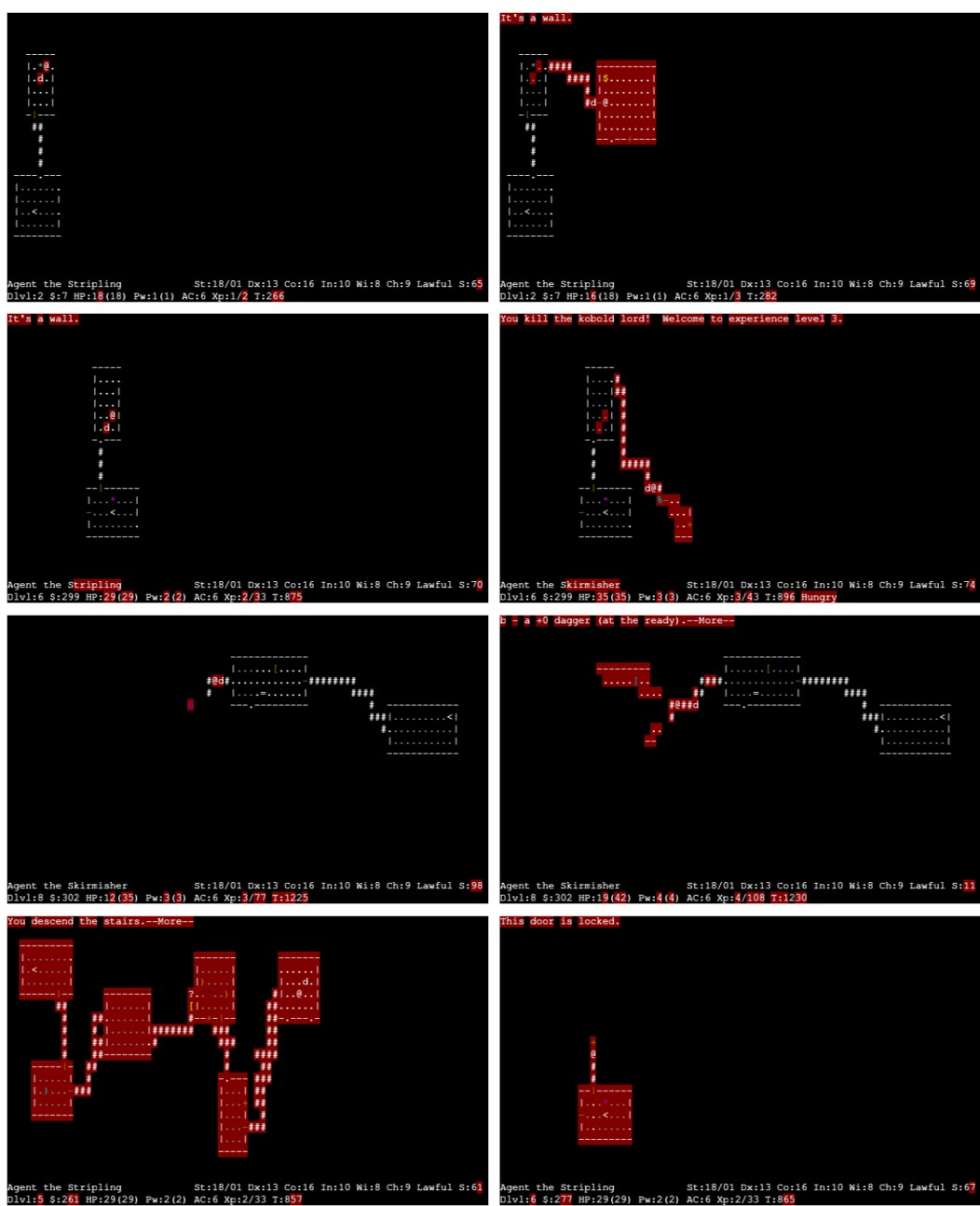

Figure 14: Visualization of the pair of frames corresponding to each peak in Fig. 13. Differences are highlighted in red: in this episode, the progress model estimates large amounts of progress for (Top) exploring the dungeon, (Middle) leveling up, and (Bottom) descending to the next dungeon level.

Fig. 17 shows a saliency map of the learned progress model a) with access to the timer, and b) without access to the timer. The magnitude of the gradient of the absolute progress estimate is represented by the brightness of the red highlight. Where explicit timing information is available, the model will focus almost exclusively on this feature. We find that this hurts the ability of ELE to explore, as even meaningless random behavior usually results in advancement of the in-game time, so we remove it from the observation in this work. Some real-world environments contain explicit timing information like this, so factoring out these features in a way that requires less domain knowledge is a promising objective for further work.

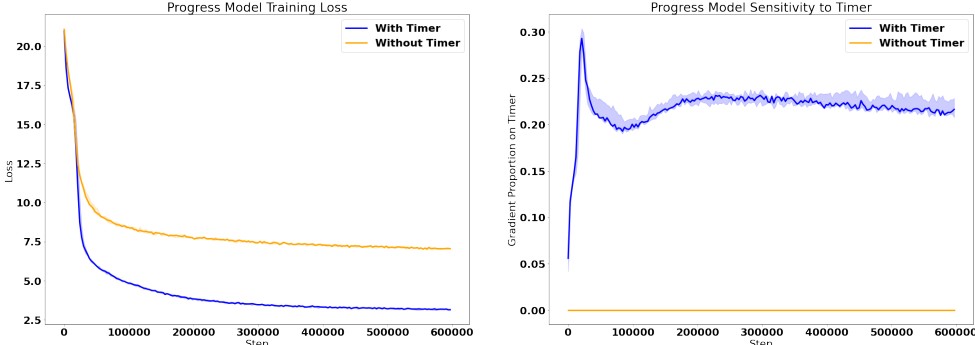

Figure 15: Left: Median learning curve for the supervised regressor over three seeds in the offline training phase. Note that with access to the timer, the loss curve reaches a much lower value, indicating that the timer feature is extremely useful for the prediction task, despite not conveying much information about progress. Right: The sensitivity of the progress model to the timer feature, as a proportion of the total gradient with respect to the input: when the model has access to the timer, it concentrates a disproportionate amount of its weights on that part of the input.

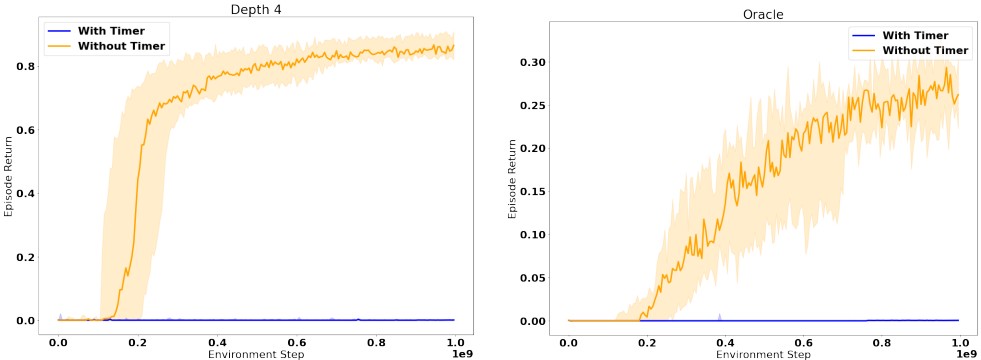

Figure 16: Episode return achieved by ELE using progress models with and without access to the timer on two of the sparse tasks: (Left) **Depth 4** and (Right) **Oracle**. The progress model with access to the timer results in poor exploration, since the estimate considers primarily the in-game time passing, rather than making meaningful progress in exploring the dungeon.

In addition to the saliency analysis, we provide an accuracy plot in Fig. 18 that demonstrates the accuracy of the trained progress model over a variety of tolerances around the target value. The models trained on each of the datasets do not differ significantly in their accuracy, and perhaps surprisingly, the accuracy itself is apparently not very high (for example, only approximately 55% of predictions are within 50% of the target value). This is likely because of the difficulty of predicting distances over very long trajectories: we sample a maximum distance of 10K timesteps during training, and for long intervals like this, there are very many different paths that could lead between two states. Nonetheless, the progress model appears to be effective as an exploration reward despite the difficulty of making accurate predictions.

### A.7 FORM AND STOCHASTICITY

Despite performing well compared to the other baselines, FORM struggles to solve the hardest tasks where ELE still succeeds. The core mechanism of FORM is to learn a predictor $p_D(s_t) \rightarrow s_{t+1}$ of state transitions under a demonstrator policy, independent of actions. This prediction task is difficult when a) the demonstration policies result in inconsistent transitions, and b) the environment dynamics are themselves highly stochastic. In our case, the demonstration policies are quite diverse: we use ten different experts who will each have different play styles, and they may not even take the same action each time they themselves encounter the same state, due to differences in mood, motivation, etc.

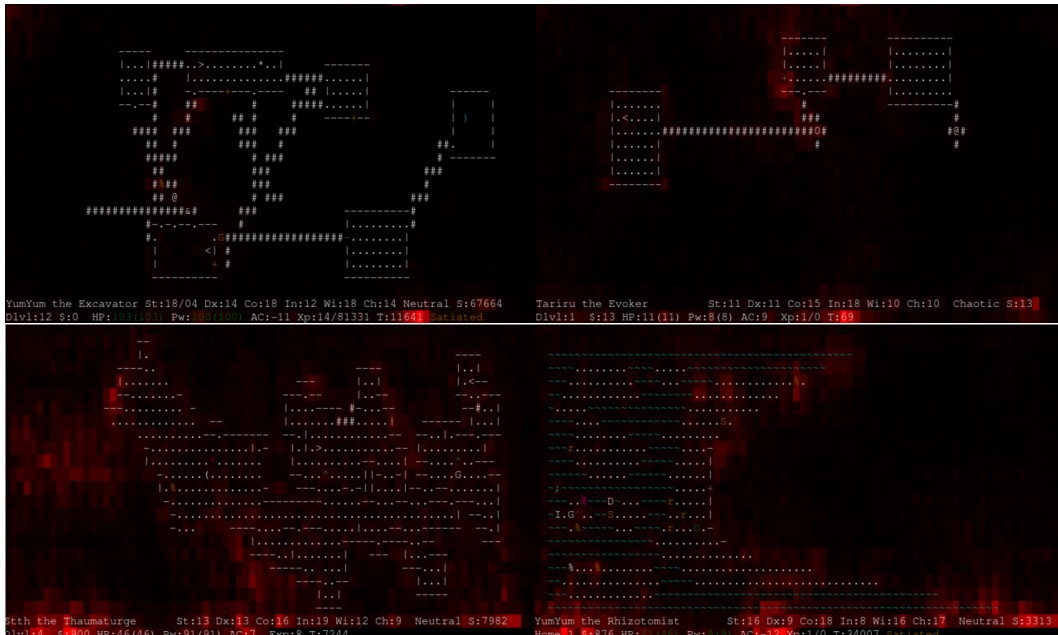

Figure 17: Saliency map for the progress model over the input calculated as $\nabla_{s_t}|f(s_{t-k},s_t)|$, (top) with and (bottom) without access to the timer observation. Top: When the timer is available, the saliency map is consistently strongest in that part of the input. Bottom: with the timer feature explicitly masked, the model is required to pay attention to more semantically interesting parts of the input such as the dungeon, and the character statistics.

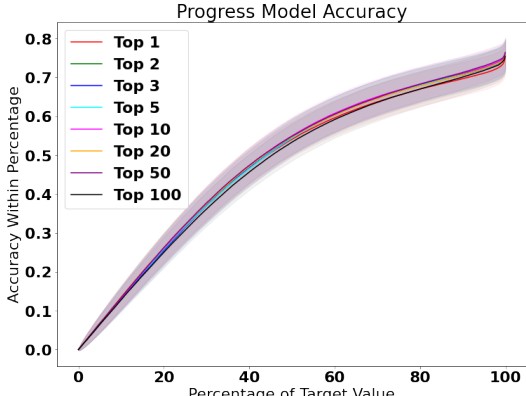

Figure 18: Accuracy of the progress model for each dataset over a variety of tolerances around the target value. Since the progress model is a regression model, it is not easy to provide an intuitive summary in a single accuracy value. Instead, we show how many predictions from the model lie within a particular percentage of the target value.

Beyond this, NetHack is stochastic and partially observed: it is not possible to predict accurately what the next dungeon level will look like, or what will be revealed behind a closed door. Both of these factors make for a challenging prediction task, and the resulting rewards from FORM may be highly inconsistent. In addition, while the FORM model is trained on a dataset of pairs $(s_t,s_{t+1})$, the ELE progress model is trained on pairs $(s_t,s_{t+k})$ for $k \in [-10000,10000]$. As a result, the effective size of the progress model dataset much larger. See Fig. 19 for example predictions and likelihood visualizations from the FORM model, demonstrating the difficulty of the prediction task.

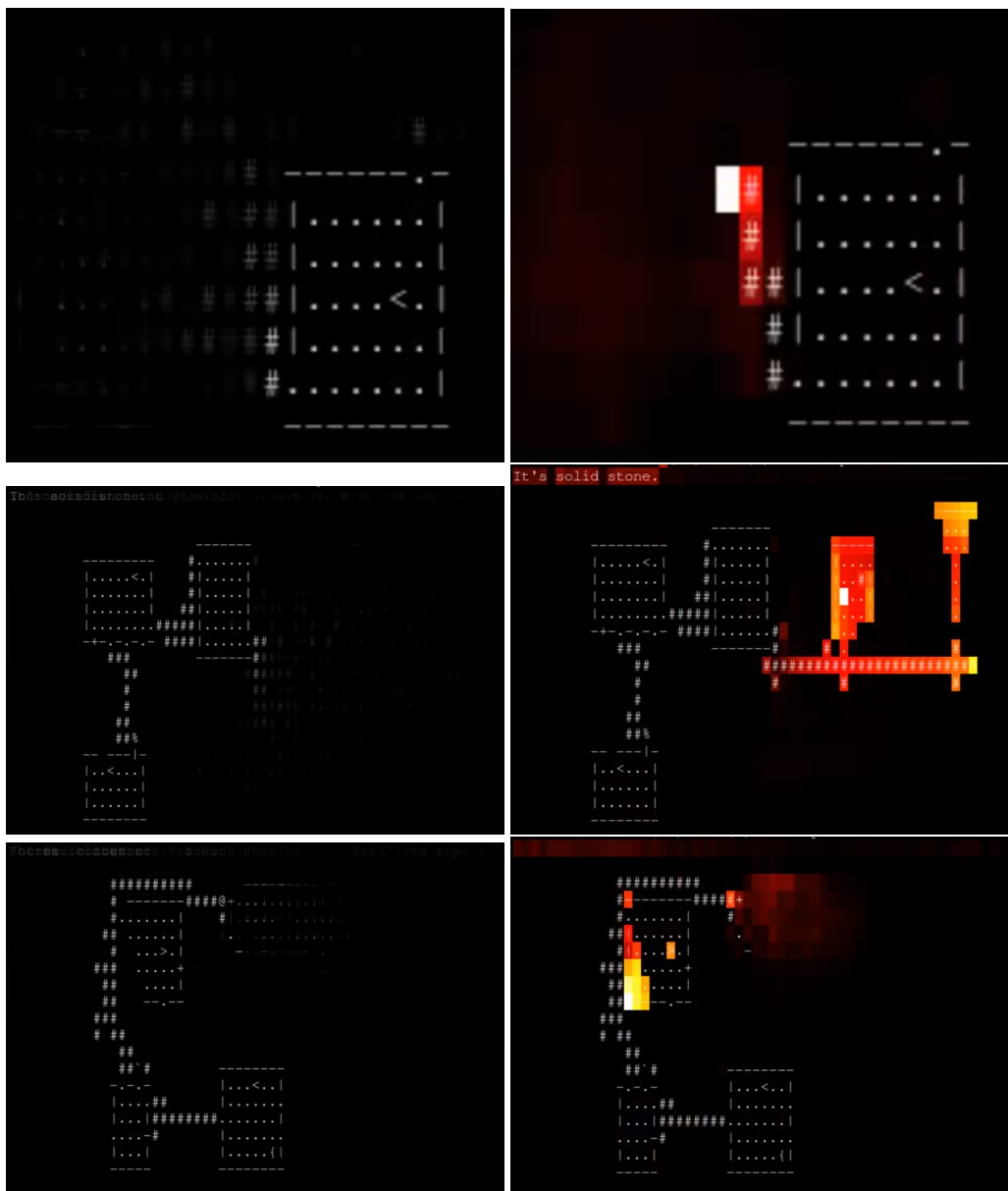

Figure 19: Predictions and likelihood visualizations from the FORM model. (Left) Average of 256 samples from the FORM model $p(s_{t+1}|s_t)$ expert demonstrations. Note that due to the inherent stochasticity of the training data, the predictions are quite fuzzy in regions of the input hidden from the player. (Right) The actual next frame $s_{t+1}$, with the surprisal under the predicted distribution highlighted by heatmap. The bright highlights indicate highly surprising elements of the observation, which are not in general possible to predict in advance due to the stochasticity in both the expert policy $p(a_t|s_t)$ and the environment dynamics $p(s_{t+1}|s_t,a_t)$.

## A.8 MOTIVATIONAL EXAMPLE DERIVATION AND CONNECTION TO STOCHASTIC PROCESSES

The mechanism at the core of ELE consists in capturing an average direction of forward progress from competent offline demonstrations. That is, knowing that the training data is coming from a reasonable policy implies that capturing its overall trend in a model and subsequently using this model for intrinsic motivation will project the RL agent onto a subspace spanned by the competent demonstrations. In this subspace, even undirected exploration can greatly improve the understanding of the agent about the world, as the deviations are concentrated along reasonable paths.

In this section, we dissect the notion of forward progress by showing how it is analogous to the concept of drift in stochastic process theory. We begin by constructing a simple 1-dimensional environment where the notion of forward progress corresponds to the drift of the stochastic process.

Suppose an agent begins an episode at the origin of the real line, and is allowed to take independent, Gaussian steps either to the left or to the right of the starting position. Moreover, the agent receives a positive reward the first time it crosses a boundary specified by $x = c$ for $c > 0$ within at most $T$ steps. This hypothetical example can be formulated as maximizing the number of boundary crossings in a Wiener process (Fu & Wu, 2010).

The core idea for solving this 1-dimensional walk on the real line is summarized as follows: if we estimate the *progress* (i.e. expected direction) of sample paths which are deemed successful according to some criteria (in this case, reaching a specific region on the line), then we can use this information to guide our learning by projecting the problem into a simpler space s.t. even random deviations of the RL agent from demonstrations can lead to improving the agent's performance.

**Lemma A.1** *Let $V(0, \Delta)$ be the value function at $t = 0$ of the policy $\vec{W}(t) = W(t) + \Delta t$, where $\{W(t) : t \in [0,T]\}$ is a Wiener process on $[0,T]$, deployed in the 1-dimensional example above. Then,*

$$\frac{c}{T} = \underset{\Delta \in \mathbb{R}^+}{\operatorname{argmax}} V(0, \Delta) \tag{2}$$

Since the angle between the expected values of the optimal policy $W(t) + \Delta t$ and $W(t)$ in the time domain is given by $\theta = \tan^{-1}(\frac{c}{T})$, $\Delta$ can be interpreted as a measure of *alignment* of the current policy with respect to the optimal one. Pre-training a progress model on data which has a high $\theta$ will provide an initial alignment, while pre-training on data closer to random has a lower $\theta$ and hence lower value. Our approach aims to use demonstrations to learn a progress model which, when used as an auxiliary reward, yields behavior which is more closely aligned to the optimal behavior than random exploration would be.

**Optimal value function depends on the drift** Let $\{W(t) : t \in [0,T]\}$ be a Wiener process on $\mathbb{R}$. Then the expected reward collected by the policy $\vec{W}(t) = W(t) + \Delta t$ (i.e. Wiener process with drift $\Delta$) over the time interval $[0,T]$ is

$$V(0; \Delta) = \mathbb{E}[\sum_{t=0}^{T} \mathbb{I}[\vec{W}(t) \geq c]] = \sum_{t=0}^{T} \mathbb{P}[W(t) + \Delta t \geq c] \mathbb{I}[W(t) + \Delta t \geq c], \tag{3}$$

where $\mathbb{P}[W(t) + \Delta t \geq c]$ can be upper-bounded using the martingale result of Robbins & Siegmund (1970)

$$\mathbb{P}[W(t) + \Delta t \geq c] \leq \mathbb{P}[\sup_{t \in [0,T]} W(t) + \Delta t \geq c] = 1 - \Phi\left(\frac{c}{\sqrt{T}} + \Delta\sqrt{T}\right) + e^{-2c\Delta}\Phi\left(\Delta\sqrt{T} - \frac{c}{\sqrt{T}}\right). \tag{4}$$

This implies that for $\Delta_1 > \Delta_2 > 0$, $V(0, \Delta_1) > V(0, \Delta_2)$, as the process $W(t)$ is symmetrically around 0. Specifically, sample paths reaching $c$ are more likely for higher values of $\Delta$. This can be shown by writing the above probability as

$$\psi(\Delta) = 1 - \Phi\left(\frac{c}{\sqrt{T}} + \Delta\sqrt{T}\right) + e^{-2c\Delta}\Phi\left(\Delta\sqrt{T} - \frac{c}{\sqrt{T}}\right). \tag{5}$$

Since $\psi$ is smooth in $\Delta$, solving the following optimization problem using Leibniz's rule

$$\Delta^* = \left\{\frac{d}{d\Delta}\psi(\Delta) = 0\right\} \tag{6}$$

yields the following relationship between path length, boundary position and drift parameter:

$$T = \frac{\sqrt{2c\Delta+1}+1}{\Delta^2} + \frac{c}{\Delta} \tag{7}$$

If, matching our 1-D experiments, we let $c = 200$ and $T = 2000$, then the estimated drift $\Delta = \frac{1}{10} + \frac{1}{10\sqrt{10}} \approx 0.1316$. Given that $\mathbb{E}[\vec{W}(t)] = \Delta t$ and $\mathbb{V}[\vec{W}(t)] = t$, the process $\vec{W}(t)$ rotates according to the angle specified by $\Delta$, and the number of boundary crossings increases when $\Delta \to \infty$.

If we wish to go further and let the agent's trajectories end withing an $\varepsilon-$ball of the goal $c$, the corresponding success probability can be computed as follows:

$$
\begin{aligned}
\mathbb{P}[c-\varepsilon \leq W(t) - \Delta t \leq c+\varepsilon] &= 1 - \mathbb{P}[W(t) - \Delta t \geq c+\varepsilon] - \mathbb{P}[W(t) - \Delta t \geq c-\varepsilon] \\
&= \Phi\left(\frac{c+\varepsilon}{\sqrt{T}} + \Delta\sqrt{T}\right) - e^{-2(c+\varepsilon)\Delta} \Phi\left(\Delta\sqrt{T} - \frac{c+\varepsilon}{\sqrt{T}}\right) \\
&\quad + \Phi\left(\frac{c-\varepsilon}{\sqrt{T}} + \Delta\sqrt{T}\right) - e^{-2(c-\varepsilon)\Delta} \Phi\left(\Delta\sqrt{T} - \frac{c-\varepsilon}{\sqrt{T}}\right) - 1
\end{aligned}
\tag{8}
$$

Maximizing the above probability as a function of $\Delta$ using Leibniz's rule yields $\Delta^* = \frac{c}{T}$, which agrees with results of Schilling & Partzsch (2012).

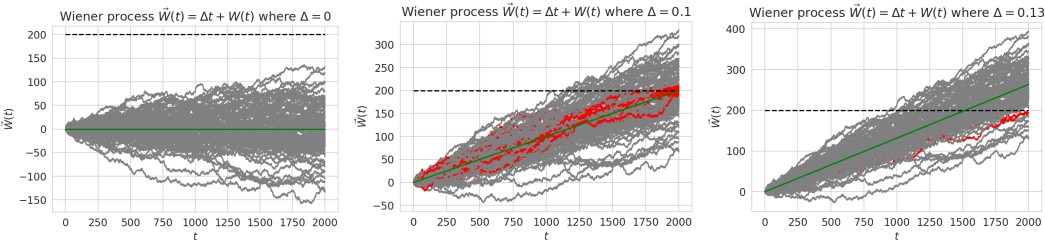

Figure 20: 100 sample paths from the Wiener process with drift $\Delta$, for various values of $\Delta$. Paths which end in the region $[190,210]$ after 2000 timesteps are highlighted in red, and the mean of $\vec{W}(t) = \Delta t$ is shown in green.

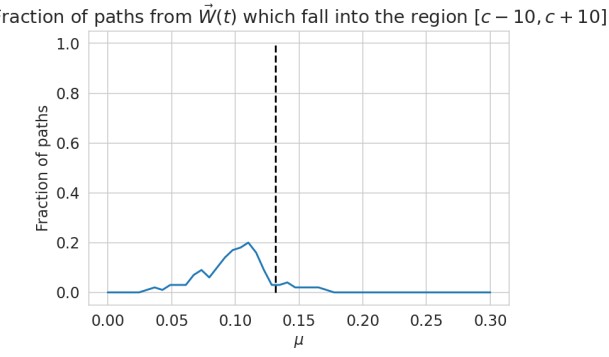

Figure 21: Fraction of sample paths which land in the region $[190,210]$ after 2000 timesteps as a function of drift parameter $\Delta$. $\Delta = 0.1$ is the value for which $\vec{W}(t)$ has the most paths landing in the goal region, and thus corresponds to maximal progress.

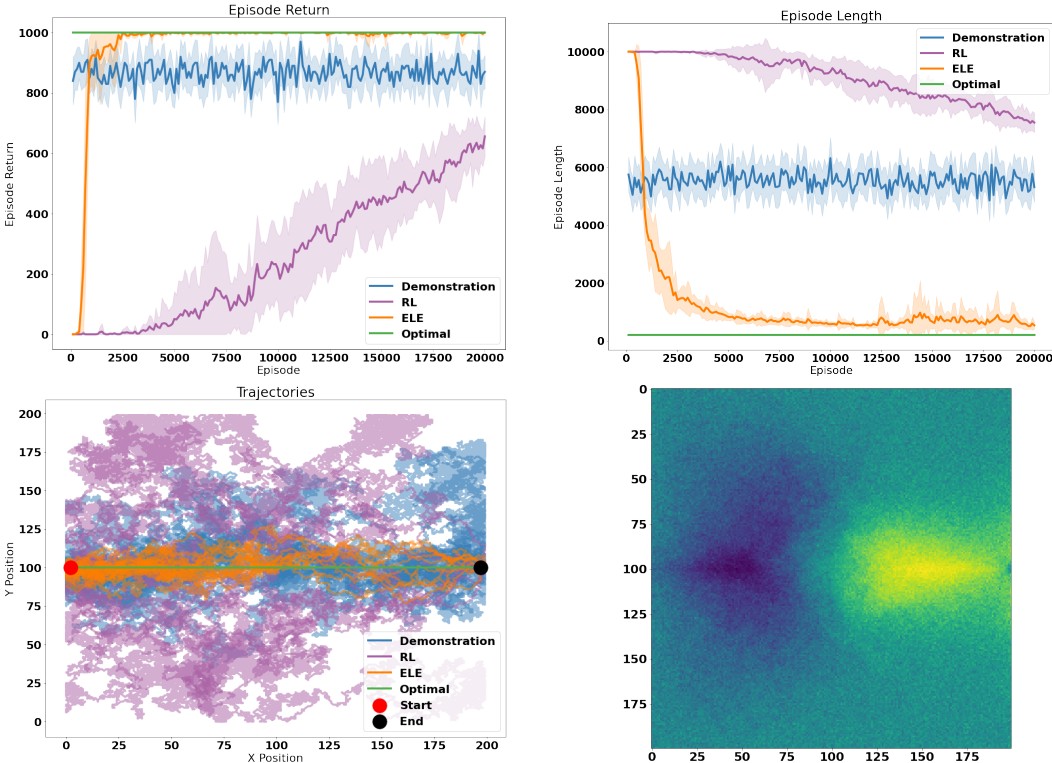

Figure 22: Toy grid environment: $200 \times 200$ square grid with tabular Q-learning agent and ELE. The agent can move in the 8 orthogonal and diagonal directions, starts at $(1,100)$ and receives a reward of 1000 and episode termination when reaching $(200,100)$. (Top Left) Episode return: the demonstrations achieve approximately 950 reward on average, while ELE receives nearly 1000, and tabular Q-learning on its own reaches approximately 700 by the time the experiment ends after 20K episodes. (Top Right) Episode length: the demonstrations and Q-learning agent take suboptimal paths to the goal, whereas tabular Q-learning + the ELE objective rapidly reaches near-optimal path lengths. (Bottom Left) 10 example trajectories from each method: ELE discovers paths that more directly approach the goal than the demonstrations that it learned from. (Bottom Right) Visualization of the ELE progress model $f^*(s_t, s_{t+k})$ for $s_t = (100,100)$ against all possible $s_{t+k}$. The model has learned that expert progress tends to be positive to the right of $s_t$ and negative to the left.

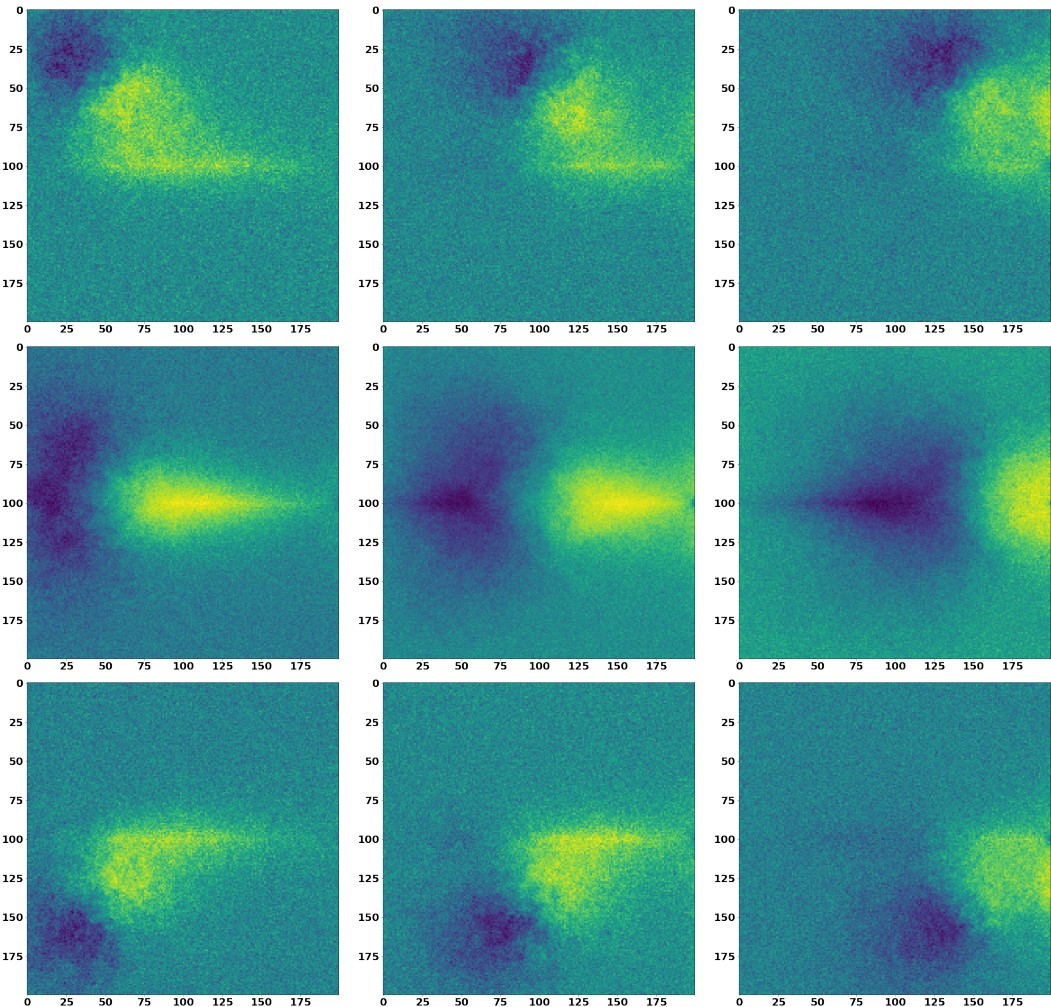

Figure 23: Progress model estimates $f^*(s_t, s_{t+k})$ from nine different states $s_t = (x, y)$ in the toy tabular environment. Rows: $x = 50$, $x = 100$, and $x = 150$. Columns: $y = 50$, $y = 100$, and $y = 150$. Progress estimates are generally high in the direction of the goal, and low in the opposite direction. Points near the $y = 100$ line are the most densely represented in the demonstration data, so the progress estimates toward the center of the grid are the most well-defined.

