# OpenReview forum: "Learning About Progress From Experts"
_ICLR.cc/2023/Conference — ICLR 2023 notable top 25%_

### Official Review · Reviewer_8ZSu · 2022-10-23

**Confidence:** 3
**Clarity, Quality, Novelty And Reproducibility:** The paper is of high quality, is clea…
**Correctness:** 4
**Technical Novelty And Significance:** 3
**Empirical Novelty And Significance:** 3
**Recommendation:** 8

**Strength And Weaknesses:**

Strengths:

Well written paper with the main idea presented very clearly and with good motivation. The method is novel (to the best of my knowledge) and clever, the experiments and baselines are comprehensive, and the toy grid-world experiment is succinct and didactic. The conclusion includes candid discussion of the limitations of this approach, which I appreciated.

Weaknesses:

I can see empirically that ELE outperforms FORM on the sparse reward tasks (figure 3), but I think some analysis of why this is the case would be helpful. Both methods get access to the same demonstration data and both are just intrinsic rewards on top of the same RL algorithm (from what I understand). So what makes ELE better than FORM?

The ablations in figure 6 shows empirically why we would want to use only demonstrations from the top 10 users, but I don't understand fundamentally why more demonstrations would hurt. For example, what about the top 10 players makes it so different from the next 10 players?

The experiments section could be improved by demonstrating the method on more than just the NetHack environment.

For which task is the right side of figure 3 plotting?

**Summary Of The Paper:**

The authors propose a method (ELE) to incorporate information from expert (observation-only) demonstrations into an RL agent by first learning an estimate of "progress" and using this estimate as an intrinsic reward. Progress in this case is defined to be the temporal distance between two states.

Results are first shown on a toy grid-world environment, where demonstrations are epsilon-greedy oracle policies (choosing the optimal action 2.5% of the time). Further experiments on 7 different tasks in the NetHack environment shows that ELE is competitive against a wide variety of baselines, and is particularly effective in sparse reward settings.

Finally, qualitative visualizations of the progress function, an episode from a trained policy, and saliency maps are presented, as well as ablations on the expert demonstrations dataset size and the "history length" hyperparameter.

**Summary Of The Review:**

The paper is well written, the idea is novel, and experiments are fairly comprehensive. Only reservations I have are in the fact that there is only one environment and the lack of analysis comparing ELE to FORM (see weaknesses section).

---

> ### Author Response · Authors · 2022-11-17
> **Response to Reviewer 8ZSu**
>
> Thanks for the suggestions and for pointing out where clarifications can be made. Specific responses below.
>
> ### Comparison of ELE and FORM:
> FORM involves learning a model $p(s_{t+1} | s_t)$ from expert demonstrations and then rewarding the agent for producing high-probability transitions under this model. This is a hard model to learn when a) the demonstration policies produce inconsistent transitions, and b) the environment dynamics are themselves highly stochastic.
>
> In our case we learn from 10 different experts with different policies, and in addition, human experts are not always consistent in their own behavior. Furthermore, the dynamics of NetHack are highly stochastic, as it is impossible in general to predict e.g. what the next level will look like, or what will be revealed behind a closed door. This means that the reward delivered to the agent is washed out by a high degree of unpredictability.
>
> Another important difference between ELE and FORM is that the training dataset for FORM consists of all pairs (s_t, s_{t+1}) in the dataset, whereas ELE is trained on pairs (s_t, s_{t+k}) where k can be as large as 10,000. This means that while the ELE progress model is faced with a challenging prediction task as well, it is trained on a much larger effective set of examples.
> We have added example predictions and likelihood visualizations for FORM to the appendix in Figure 19, to support the claim about stochasticity.
>
> ### Datasets of different sizes:
> We investigate why the performance drops when we use data from more players in the appendix A.2. Reiterating, we consider two primary analyses. Firstly, all of our sub-tasks (although very complex and sparse for our agents) still correspond to the early game of NetHack. We see in Fig 9 that the Top10 dataset, which is the best performing dataset in our experiments, also includes the highest proportion of gameplay in these initial dungeon and experience levels
>
> Secondly, in Figure 10, we quantify the coverage of different datasets by measuring the KL divergence between the distribution of particular features of each dataset against all others, and we observe that the Top10 dataset has the best overall similarity to the others, indicating that it has the most general coverage.
>
> And finally, although there appears to be a significant difference between the performance of ELE using the different datasets, note that we are showing the performance on the hardest task, where none of the baselines achieve any reward: in other words, the differences between datasets are small relative to the differences between approaches.
>
> ### Other environments:
> While we agree with the reviewer that the high level idea of ELE is general and can be applied to many other environments, we believe Nethack is in many ways one of the most complex simulation environments available today, where state-of-the-art agents like Muesli and MuZero fail to solve many of these tasks. Combined with the availability of the large unlabeled dataset, these reasons make NetHack an ideal choice for evaluating our approach.
>
> As mentioned in our discussion of the limitations of this work, some non-trivial effort and care is required while extending ELE to other domains, for example those having explicit notions of progress like a “timer” in the observations, or highly cyclic dynamics (both features are common in Atari games for example). We believe this work illustrates that learning a monotonic function from demonstrations is possible and useful in a complex environment like Nethack, and extending this idea further to other complex domains with appropriate modifications is a useful effort for future work.
>
> ### Figure 3:
> The right side of Figure 3 is a Mann-Whitney U-Test aggregated over samples of all seeds and tasks. This corresponds to the analysis recommended in Agarwal et al. 2021 with sample_size = n_seeds * n_tasks = 35.

---

> > ### Comment · Reviewer_8ZSu · 2022-11-30
> > **Reply to Authors**
> >
> > Dear Authors,
> >
> > Thank you for the rebuttal. Upon reading the other reviewers' comments and corresponding rebuttals I've decided to update my score to 8. This is a well-written paper with a well-executed idea and good motivation, and the rebuttal has adequately addressed my concerns.

---

### Official Review · Reviewer_iUvn · 2022-10-25

**Confidence:** 3
**Correctness:** 3
**Technical Novelty And Significance:** 3
**Empirical Novelty And Significance:** 3
**Recommendation:** 6

**Clarity, Quality, Novelty And Reproducibility:**

The presentation of the method is clear. Some baseline implementation can be further clarified.

**Strength And Weaknesses:**

The paper presented a simple but promising approach to learning intrinsic rewards for exploration. The model can be learned from unlabelled offline data, which contain human expert demonstrations on possibly irrelevant tasks. The model outperforms several baselines on tasks in the NetHack environment.

Overall I think the paper presentation is clear, and the comparisons are sufficient. There are a few points for potential improvements.

1. Is there any way that we can test the accuracy of the pretrained distance prediction model? How does that really translate into performance?
2. I am not familiar with the NetHack environment. a. Is the map fixed? b. Do the tested tasks require handling partial observability? I am asking because If the map is not fixed how can you model learn the distance function between different "levels?"
3. How similar are the demonstrations to the testing tasks? Are expert demonstrations exactly for the task (e.g., oracle)?
4. My understanding of the answer to Q3 is no. But then, I don't understand how the baseline BCO works.
5. Also, while I understand that there will be a lot of "noise" in such a large-scale dataset, Figure 6 makes me very worried: it seems that the authors have just used data from 10 players. Do you have any guesses about why model performance drops when you have more data? Is it because the larger dataset requires a larger model? Or is it because of the "noises?" If so, what is exactly the "noises?" Or is it an issue with the proposed method? (e.g., the proposed objective is too hard to be learned from Internet data).
6. The authors should consider comparing with works on "subgoal generation." For example,
Czechowski, Konrad, et al. "Subgoal search for complex reasoning tasks." Advances in Neural Information Processing Systems 34 (2021): 624-638.
7. Is the state-only setting a real constraint of the environment/dataset? That is, is there any way to obtain large-scale data with actions?
8. How will the model compare with other exploration strategies, such as Go-Explore (Ecoffet et al., 2019)?


**Summary Of The Paper:**

This paper presents an approach for improving agent exploration in reinforcement learning tasks using a learned distance model from an unlabelled dataset of human plays. The training paradigm has two steps. First, the authors train a distance predictor for each pair of states. The distance is used to help agents to explore as an intrinsic reward signal.

**Summary Of The Review:**

Overall the paper presented an interesting idea and promising results.

---

> ### Author Response · Authors · 2022-11-17
> **Response to Reviewer iUvn**
>
> Thank you for the clarifying questions and suggestions. Specific responses follow.
>
> ### Accuracy:
> Besides the saliency analysis already in the appendix, we have added an accuracy plot (Figure 18). We plot the proportion of predictions that are within a particular percentage of the target value, for values from 0% (perfectly accurate) to 100% (off by an absolute value equal to the target). We find that the progress models achieve approximately 55% accuracy within a threshold of 50% of the target value.
>
> We note that this accuracy is not particularly high, because it is hard, in general, to predict long timescales accurately: we sample distances up to 10,000 timesteps, so precise predictions will often be impossible.
> Time did not permit conducting an experiment with early stopping to evaluate how well ELE works with a partially trained model. Nonetheless, we find that the progress model is quite effective, perhaps because it is used over shorter timescales during the RL phase.
>
> ### Nethack map:
> The nethack map is procedurally generated and is highly stochastic. However, some aspects of the observations reliably indicate progress in the game. ELE’s progress model captures many of these subtle indicators from a diverse set of demonstrations, despite the stochasticity: see Figure 5 (bottom) and Figure 17 showing the sensitivity of the progress model to features including character and dungeon level, score, gold, and armor class.
>
> ### Similarity of demonstrations to testing tasks:
> The demonstrations are human gameplay recorded over the past 14 years, who usually aim to win the game. These demonstrations are somewhat aligned with the benchmark tasks proposed here: the sparse tasks represent milestones that are likely to occur over the course of winning the game, but the players were not aiming to solve these specific tasks. We will clarify this in the manuscript.
>
> ### How BCO works:
> Although the demonstrations are not obtained directly for the benchmark tasks, the BCO loss acts as a policy regularizer toward the types of behavior that are generally useful for the game. This is why we add the extrinsic reward as an objective to all baselines: the purpose of the imitation losses is to encourage exploration.
>
> ### Datasets of different sizes:
> We investigate why the performance drops when we use data from more players in the appendix A.2. Reiterating, we consider two primary analyses. Firstly, all of our sub-tasks (although very complex and sparse for our agents) still correspond to the early game of NetHack. We see in Fig 9 that the Top10 dataset, which is the best performing dataset in our experiments, also includes the highest proportion of gameplay in these initial dungeon and experience levels
>
> Secondly, in Figure 10, we quantify the coverage of different datasets by measuring the KL divergence between the distribution of particular features of each dataset against all others, and we observe that the Top10 dataset has the best overall similarity to the others, indicating that it has the most general coverage.
>
> And finally, although there appears to be a significant difference between the performance of ELE using the different datasets, note that we are showing the performance on the hardest task, where none of the baselines achieve any reward: in other words, the differences between datasets are small relative to the differences between approaches.
>
> ### Comparison with Sub-goal generation:
> Thanks for the suggestion. We agree that sub-goal generation could be another point of attack to this problem and is a useful direction for future work.
>
> ### State-only setting:
> The demonstrations used in this work correspond to screen recordings of human gameplay over a period of 14 years. It would be possible for the owners of the server to implement action recording for games going forward, but it is not generally possible to infer all actions from state-only observations for the purpose of labeling the existing dataset, and it would be prohibitively expensive to record a similar amount of data with actions from scratch.
>
> Moreover, action-free imitation is a well-studied setting in the literature, and the prospect of imitation from observations alone is attractive in situations where obtaining actions could be impossible or expensive. This is the setting we aim to address, and we will clarify this motivation in the manuscript.
>
> ### Comparison with Go-Explore:
> As described in Sec 2 (page 2, last paragraph), approaches like Go-Explore assume that the environment can be reset to arbitrary states and gameplay resumed from there. Follow-up work to Go-Explore relaxes this assumption, but still requires that arbitrary states can generally be reached in every episode. These assumptions do not hold in many externally defined environments like NetHack, where the execution environment is a black box and the structure of the world is procedurally generated.

---

> > ### Comment · Reviewer_iUvn · 2022-11-24
> > **Thanks for the response**
> >
> > Thank you for the response. I vote for acceptance of the paper. I hope the authors can include these clarifications about the environment and baselines in their revision. I still think comparison with other sub-goal based methods would be an important and interesting comparison to be included in **this** paper.

---

### Official Review · Reviewer_hMhc · 2022-10-28

**Confidence:** 3
**Correctness:** 4
**Technical Novelty And Significance:** 2
**Empirical Novelty And Significance:** 4
**Recommendation:** 8

**Clarity, Quality, Novelty And Reproducibility:**

* Clarity: Clarity was good overall, though there were a few points which could benefit from additional clarity:
   * It would be better to define the setting of interest as long-horizon, sparse reward with limited expert demonstrations ("action-free imitation learning"), instead of just mentioning that the algorithm has the benefit of not requiring full expert demonstration info. The reason I propose this modification is because the paper appears to focus on and test exclusively this setting, rather than potentially broader applications of the idea of measuring progress.

* Quality: The paper was of good quality.

* Novelty: The main ideas in the paper appear to be novel. The related work section seemed good, but I am not extremely familiar with the literation on action-free imitation learning.

* Reproducibility: The reproducibility is great - experiment details are provided and they will release the dataset.

**Strength And Weaknesses:**

* Strengths:
   * The paper considers a highly relevant and interesting setting involving expert information, long timescales, and sparse reward signals.
   * The paper proposes the interesting (and apparently novel) idea of estimating a temporal progress function (while controlling for removing trivial signals from which progress can be deduced), and finds that with some specific choices of how the progress function estimation is implemented and used as an auxillary reward, that it is quite beneficial. This seems like quite a nice simple principle to uncover if it was previously unknown.
   * The paper provides extensive experiments and comparisons on their domain of choice (which seems to be a reasonable stand-in for any setting with incomplete expert demonstrations), and they examine performance on several tasks with uncertainty estimates for the performance of their algorithm beating existing algorithms.
   * An open-source curated dataset will be released for the NetHack environment, which may prove useful for further work on long timescale with sparse reward and incomplete demonstration information.
   * Some useful empirical choices in the estimation of the progress function were uncovered as a result of this work: shorter timescale prediction of progress performs better than not, and log scaling during the training of the progress function was also useful.
   * A fairly thorough related work section is provided.

* Weaknesses:
   * It would be very useful to additionally examine a setting where the expert demonstrations are not incomplete; in such settings (still with long time horizons and sparse rewards), how does ELE fair against standard imitation learning algorithms like Behavior Cloning and DAgger (and more recent variants)? That would help disambiguate whether ELE is useful only in the context of missing demonstration information, or whether the idea is more broadly useful even when you have full expert demonstrations.
   * It would be nice to provide a clearer justification and understanding (empirical or theoretical) of why estimating the progress function is helpful; likewise with the insight that shorter time horizons are more useful to train on.
   * Only one underlying RL algorithm was tested (Muesli) unless I missed something -- it would be quite valuable to understand whether the underlying RL algorithm being used affects the utility of the augmented reward signal.


**Summary Of The Paper:**

This paper studies the question of how to leverage expert demonstrations when explicit actions are not present in RL environments which have long timescales and sparse rewards, but for which there exists a notion of monotone increasing progress over time. The central idea of this paper is to use the (incomplete) expert demonstration data to estimate a function which predicts the temporal distance between two states. Then, this function is used to define a weighted auxillary reward which measures the progress between the current state and the state $k$ steps in the past; $k$ is a hyper-parameter which is tuned. RL algorithms without this additional signal are then compared to RL algorithms which use this modified reward function, this meta-algorithm is referred to as ELE, and experiments are conducted both in a toy gridworld setting and in the first 1M steps of the game NetHack, a challenging environment with long timescales and sparse rewards that has been previously studied for which incomplete expert demonstrations are available. Compared to other learning algorithms which work over the same information, ELE outperforms (in some cases, quite significantly) the other methods on a variety of sparse signal long-horizon tasks. A curated dataset of expert demonstrations on Nethack (subsampled from an existing dataset) is also planned to be released.

**Summary Of The Review:**

Overall, the paper seems interesting and clear. I would lean towards accepting the paper.

====== POST-REBUTTAL =======

Having read the other reviews and responses, I remain convinced that the paper should be accepted.

---

> ### Author Response · Authors · 2022-11-17
> **Response to Reviewer hMhc**
>
> Thanks for the insightful comments, suggestions, and the review. Responses to specific suggestions follow.
>
> ### Comparisons with standard imitation algorithms having access to actions:
> We would like to emphasize that in general, action-labeled datasets are far harder to obtain. In many cases, large amounts of observation-only data is available as a result of passive recordings (e.g. YouTube data, Twitch streams, etc.). By contrast, action-labeled data often requires dedicated data gathering with action capture, and in the case of the NetHack dataset which consists of games recorded over a period of 14 years, matching this quantity of data would be prohibitively expensive. In particular, assuming the steps per second we report in Table 5, it would take just over 11 years of continuous wall-clock time for a single human of our skill level to record the amount of data we used for our main results.
>
> This work focuses exclusively on scenarios like these, involving data available “in the wild”. This setting is commonly studied and motivated in many recent imitation learning works, and we will clarify the setting of interest in the manuscript. We believe a detailed investigation of progress models with access to actions would be significant work in itself and we view this as an important future direction.
>
> ### Why the progress model is useful:
> The main idea of using a progress model is that instead of matching consecutive state transitions (as done in GaiFo, BCO, FORM etc) which captures only the shortest timescale, our progress model encourages higher order temporal progress over multiple frames.
>
> 1) Our visualizations in Figure 5 indicate that peak progress corresponds to a new room being opened and the saliency visualizations (bottom plot) confirm that this model is sensitive to level, score, and so on, which are semantically meaningful indicators of progress in the game.
>
> 2) We further try to empirically understand the progress model by investigating a toy example in Figure 2. The left side of the figure shows that even when demonstrations are highly sub-optimal, ELE results in much shorter paths, without attempting to match every action in the demonstration trajectories. Figure 2 (right) visualizes the progress model on state (100,100) in the center, and there is a clearly interpretable smooth flow toward the goal, indicating that the model has captured the non-local notion of progress that we intend from our approach.
>
> ### Other RL algorithms:
> In this work, we implemented ELE and other imitation learning baselines on top of Muesli, one of the strongest reinforcement learning agents in Nethack. The underlying agent outperforms published results on the Scout and Score tasks by a large margin (1400 vs 1000 on Score, and 3425 vs 1750 on Scout) despite playing a harder version of the game with randomized characters and the full action space [6]. While ELE and the other imitation algorithms could be used with alternative agents like R2D2 or IMPALA, we found in our preliminary experiments that these other algorithms perform quite poorly on NetHack compared to Muesli, especially for the sparse tasks we examine in this work. The method does not rely on any particular details of the underlying algorithm, and building on a single strong underlying agent is also common practice in other recent works.

---

### Decision · Program_Chairs · 2023-01-20

**Decision:**

Accept: notable-top-25%

**Justification For Why Not Higher Score:**

I think this paper is really good and presents a novel and clever idea. Reviewers strongly supported this paper.
I have provided a Spotlight rating since it would be have really great if the paper demonstrated this approach on either another challenging game task or perhaps an embodied/robotics task (simulation or real). In the absence of that, I am voting for a spotlight. I am on the fence though, and would be ok if the paper received an oral presentation.

**Justification For Why Not Lower Score:**

N/A

**Metareview: Summary, Strengths And Weaknesses:**

This paper presents a method to use incomplete expert demonstrations to train progress functions for the corresponding task. Such progress functions can serve as intermediate rewards to train RL based agents for tasks that do come with only sparse rewards. The paper demonstrates this method on a gridworld and on NetHack.

Three reviewers provided reviews for the paper and were consistently positive. They found the method to be novel and interesting, experiments to be extensive and the paper to be clear. The reviewers had a few low level questions for the authors, which were well addressed in the rebuttal. Reviewers were pleased with the answers and improvements and raised their scores to provide final scores of 8, 8, 6 and unanimously upported accepting the paper. Given the reviews, rebuttal, scores and my reading of the paper, I recommending an accept.

**Note From Pc:**

if the above contains the word "oral" or "spotlight" please see: "oral" presentation means -> notable-top-5% and "spotlight" means -> notable-top-25%. As stated in our emails, we are disassociating presentation type from AC recommendations

**Summary Of Ac-Reviewer Meeting:**

N/A